# Characteristics and rarity of the strong 1940s westerly wind event over the Amundsen Sea, West Antarctica

Gemma K. O'Connor[1], Paul R. Holland[2], Eric J. Steig[1,3], Pierre Dutrieux[2], Gregory J. Hakim[3]

[1]Department of Earth and Space Sciences, University of Washington, Seattle, WA, USA
[2]British Antarctic Survey, Cambridge, UK
[3]Department of Atmospheric Sciences, University of Washington, Seattle, WA, USA

*Correspondence to:* Gemma K. O'Connor (goconnor@uw.edu)

**Abstract.** Glaciers in the Amundsen Sea Embayment of West Antarctica are rapidly retreating and contributing to sea level rise. Ice loss is occurring primarily via exposure to warm ocean water, which varies in response to local wind variability. There is evidence that retreat was initiated in the mid-20th century, but the perturbation that may have triggered retreat remains unknown. A leading hypothesis is that large pressure and wind anomalies in the 1940s drove exceptionally strong oceanic ice-shelf melting. However, the characteristics, drivers, and rarity of the atmospheric event remain poorly constrained. We investigate the 1940s atmospheric event using paleoclimate reconstructions and climate model simulations. The reconstructions show that large westerly wind anomalies occurred from ~1938-1942, a combined response to the very large El Niño event from 1940-1942 and other variability beginning years earlier. Climate model simulations provide evidence that events of similar magnitude and duration may occur tens to hundreds of times per 10kyr of internal climate variability (~0.2 to 2.5 occurrences per century). Our results suggest that the 1940s westerly event is unlikely to have been exceptional enough to be the sole explanation for the initiation of Amundsen Sea glacier retreat. Additional factors are likely needed to explain the onset of retreat in West Antarctica, such as naturally arising variability in ocean conditions prior to the 1940s or anthropogenically driven trends since the 1940s.

## 1 Introduction

Glaciers in the Amundsen Sea Embayment (ASE) of West Antarctica are rapidly retreating, contributing significantly to global sea level rise. Ice loss is occurring via ocean-driven melting of the ice shelves that buttress the glaciers, causing the glaciers to retreat at an accelerating rate (Pritchard et al., 2012; Shepherd et al., 2019; Smith et al., 2020). There is evidence that these glaciers have been relatively stable for the last ~10,000 years (Larter et al., 2014), which implies that a change in ocean circulation, and a corresponding increase in heat delivery to the ice shelves, likely occurred to trigger the current stage of retreat. Direct observations of glaciological, oceanic, and atmospheric conditions in the ASE from recent decades show that ice melt rates are sensitive to short-term (i.e., seasonal to interannual) changes in ocean forcing and surface mass balance (Shepherd et al., 2002; Dutrieux et al., 2014; Jenkins et al., 2018; Alley et al., 2021; Wahlin et al., 2021). However, the current stage of glacier retreat is dominated by underlying ice/ocean feedbacks (Joughin et al., 2014; Holland et al., 2023). The brevity of instrumental data in this

region (available from 1979 or later) makes it difficult to assess the historical trigger that initiated the current stage of glacier retreat.

There is evidence for a link between the poleward transport of the warm Circumpolar Deep Water (CDW) that is melting the ice shelves and westerly winds over the ASE continental shelf break region (Thoma et al., 2008; Dutrieux et al., 2014; Jenkins et al., 2016). Previous studies based on climate models and regional ocean models suggest that a trend toward stronger westerly conditions over the 20$^{th}$ century would increase the amount of warm CDW that is transported toward the glaciers, triggering increased ice-shelf melt (Holland et al., 2019; Naughten et al., 2022).

However, proxy-constrained reconstructions of wind conditions in the ASE show no evidence of a westerly trend at the shelf break; instead, the average trend over the 20$^{th}$ century in this region has been weakly easterly, associated with a deepening of the Amundsen Sea Low (O'Connor et al., 2021a; Dalaiden et al., 2021; Holland et al., 2022).

Previous studies have suggested that, rather than a trend, an extremely large climate anomaly around 1940 is a

candidate event for triggering glacier retreat. Schneider and Steig (2008) used a network of ice core data to show that a warming occurred in West Antarctica from ~1936 to 1945, likely accompanied by a high sea level pressure anomaly over the Amundsen Sea. They attributed this to the very strong 1940-42 El Niño event because of the similar timing and the known teleconnection between atmospheric circulation in the Amundsen Sea and tropical Pacific climate variability (Lachlan-Cope and Connolley, 2006; Ding et al., 2011). Steig et al. (2012, 2013) showed that similar West

Antarctic warming events are consistent with enhanced westerlies at the shelf break and suggested that the 1940 warming was associated with enhanced westerlies and thus enhanced transport of warm CDW. These studies are limited by a lack of reliable pressure and wind data in this region before the 1980s, leaving large uncertainties in the atmospheric circulation patterns over the Amundsen Sea around 1940. There is additional evidence from sediment cores that show the 1940s is a notable period for the glaciers, as Pine Island Glacier may have started to retreat at this

time (Smith et al., 2017). Together, these results suggest that glacier retreat was triggered by an exceptional wind-driven anomaly in ocean circulation and ice-shelf melting that occurred around 1940.

Until recently, evaluation of the idea that atmospheric circulation changes in the 1940s were a critical trigger for glacier change was hindered by a lack of observational constraints in the ASE region. Recent developments in proxy

data assimilation have enabled more reliable reconstructions of pressure and winds in the Amundsen Sea throughout the full 20$^{th}$ century (O'Connor et al., 2021a, Dalaiden et al., 2021). Holland et al. (2022) used reconstructed pressure and winds from O'Connor et al. (2021a) to evaluate the relative influences of anthropogenic forcing and internal climate variability on trends in the ASE. They found that internal variability may have played a large role in opposing the effects of anthropogenic forcing. Rather than a century-scale trend suggested by poorly constrained climate model

simulations, a prominent feature of the proxy reconstructions over the Amundsen Sea is a large westerly anomaly at the shelf break during the 1940s, consistent with the suggestion that this may have been a key event for initiating glacier retreat.

In this study, we use annually resolved proxy reconstructions and climate model simulations of surface pressure and winds to further investigate the hypothesis that a large atmospheric event in the ASE around 1940 forced the ocean-induced changes responsible for triggering glacier retreat. Specifically, we investigate the significance of the atmospheric component of the hypothesis. Was the atmospheric event exceedingly rare, providing a possible explanation for the initiation of glacier retreat after millennia of stability? Or was it a relatively common event, perhaps superimposed on other glaciological or oceanic conditions favorable for initiating retreat? We note that the 1940s ASE atmospheric event is likely also important for influencing other important components of the local ice/ocean system, such as carbon uptake, upwelling of nutrients, and sea ice extent (e.g., Stammerjohn et al., 2015; Yager et al., 2016;), but the primary motivation of this study is to investigate the significance of the event as a potential atmospheric trigger of glacier retreat.

To investigate the significance of the atmospheric event, we first evaluate the timing and magnitude of the pressure and wind anomalies in the ASE around 1940, using two different proxy reconstructions to assess uncertainty. Next, we use proxy data assimilation to generate new reconstructions with certain proxy types withheld, allowing us to evaluate potential drivers associated with the reconstructed ASE anomalies. We also evaluate historical "pacemaker" simulations constrained by instrumental sea surface temperatures from the Pacific, Indian, and Atlantic Oceans to investigate additional sources of variability associated with the 1940s ASE anomalies. Finally, we evaluate whether the anomalies are rare in the context of 10,000 years of simulated internal climate variability, providing an estimate for the exceptionality of the 1940s atmospheric event.

## 2 Methods

### 2.1 Paleoclimate reconstructions

We use paleoclimate reconstructions to characterize anomalies in sea level pressure (SLP) and surface wind ($U_S$) in the ASE shelf break region (70-72°S, 102-115°W; location shown in Figs. 1, 3; mean SLP and $U_S$ fields from modern instrumental data are shown in Figure 1). Instrumental reanalysis products such as 20CR (Slivinski et al., 2019) and the Fogt et al. (2019) pressure reconstruction include the 1940s but are considered unreliable in the Amundsen Sea region given the sparsity of relevant instrumental observations (Fogt et al., 2019; Wohland et al., 2019; Holland et al., 2022). We therefore focus only on reconstructions that use proxy data, as consistent coverage from relevant regions throughout the full 20[th] century is available. This includes multiple, precisely dated ice core records from the relevant West Antarctic region (e.g., Steig et al., 2013; Thomas et al., 2017; PAGES2k consortium 2017) and coral records from the critical tropical Pacific region (e.g., Cobb 2002; Sanchez et al., 2020; 2021; PAGES2k consortium 2017).

We use two paleoclimate reconstructions in this study. The first is a proxy-based reconstruction from O'Connor et al. (2021a), which we refer to as the "natural-prior reconstruction". This gridded 20[th] century reconstruction (1° spatial resolution, available from 1900 to 2005) spans the Southern Ocean and includes data from a global database of proxy

records (see Fig. 1 and O'Connor et al., 2021a for proxy locations). The reconstruction was generated using the Community Earth System Model (CESM) 1 Last Millennium climate model simulation (Brady et al. 2019), without anthropogenic forcing, as the data assimilation prior. The prior is used as the initial estimate of the climate state and provides the estimate of climate covariance patterns. Four other proxy reconstructions of 20th century atmospheric

circulation are available in this region, which use different techniques and proxy databases (Dalaiden et al., 2021) or different climate model priors (O'Connor et al., 2021a), but produce very similar results. The natural-prior reconstruction from O'Connor et al. (2021a) shows the greatest skill, especially for zonal winds, in the key ASE continental shelf break region (Figs. 1, A1). Skill is based on correlation and coefficient of efficiency (a measure of signal amplitude and bias) relative to modern ERA5 reanalysis winds (Hersbach et al., 2020; data are constrained by

satellite infrared sounding observations) for the overlapping period of 1979 to 2005.

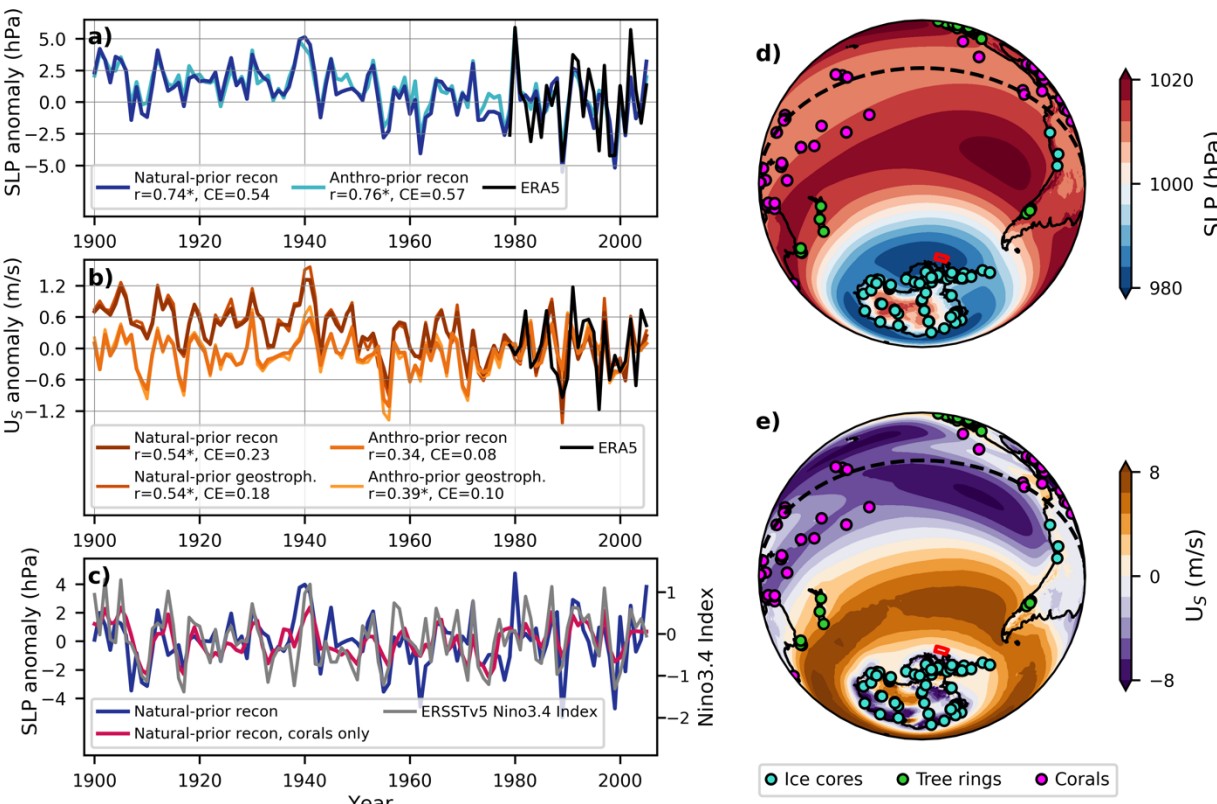

**Figure 1.** (a) Reconstructed SLP anomalies in the natural-prior reconstruction ensemble mean, anthro-prior reconstruction ensemble mean, and ERA5 averaged over the continental shelf break box (location shown by red box in panels d and e, and in black box in Fig. 3). Correlation (r) and coefficient of efficiency (CE; values > 0 are skillful)

values between each reconstruction and ERA5 from 1979 to 2005 are shown in the legend. An asterisk is given next to the correlation value if it is significant with 95% confidence. (b) Same as in (a), but for $U_S$. Geostrophic winds generated using the SLP reconstructions are also shown. (c) SLP in the natural-prior reconstruction with all proxies assimilated and with only coral records assimilated. Also shown is the Nino3.4 Index from ERSSTv5. All timeseries in panel (c) are detrended. The anomaly reference period for panels a-c is 1979-2005. (d) Mean SLP in ERA5 from

1979-2020. Locations of the proxy data used in the reconstructions are shown by the colored markers (see O'Connor et al., 2021a for complete map of proxy data). The equator is shown by the dashed line. (e) Same is in (d), but for $U_S$.

While the natural-prior reconstruction shows the greatest agreement with ERA5 in terms of interannual variability, the relatively brief 27-year verification period precludes evaluation of the reliability of multi-decadal or century-scale trends in the reconstructions. Evaluation of trends is particularly important for analyzing the 1940s, as the magnitude of the 1940s wind anomalies is sensitive to the reconstructed trends (Figs. 1, A1). The reconstructions generated with naturally forced climate model priors show easterly trends, while those that include anthropogenic forcing in the climate model prior show negligible trends – consistent with previous studies that show that greenhouse gases and ozone induce westerly trends (Arblaster and Meehl, 2006; Thompson et al., 2011; Bracegirdle et al., 2014; 2020; Goyal et al., 2021) opposing the naturally induced easterly trends (Holland et al., 2022). As a result of the uncertainty associated with the reconstructed trends, we also evaluate the O'Connor et al. (2021a) reconstruction that uses a climate-model prior with historical anthropogenic forcing; we refer to this as the "anthro-prior reconstruction". The prior for this reconstruction comes from the Community Earth System Model v1 (CESM1) tropical Pacific "pacemaker" ensemble of 20 simulations (Schneider and Deser, 2018). The anthro-prior reconstruction shows the lowest magnitude of zonal wind anomalies during the 1940s (Figs. 1, A1). Together with the natural-prior reconstruction, this makes it ideal for evaluating the full range of uncertainties among the available reconstructions.

The SLP reconstructions are more skillful than the wind reconstructions (as evaluated against ERA5), so we consider both the winds reconstructed by O'Connor et al. (2021a) and geostrophic winds calculated using the SLP reconstructions, following the approach used in Holland et al. (2022). This is especially useful for the anthro-prior reconstruction, which shows increased skill when we use geostrophic winds rather than the directly reconstructed winds (Holland et al., 2022; Fig. 1).

Specifics on how the two reconstructions were generated can be found in O'Connor et al. (2021a), but we emphasize two key qualities: (1) the climate-model prior is used as an "offline" prior (the same initial guess is used every year), so all of the variability in the reconstruction is generated from the proxy data, rather than from the climate model simulation; (2) the influence of each proxy record on the reconstructed climate in a given location depends on the covariance structures in the climate model prior (this explains the differences in the two O'Connor et al. (2021a) reconstructions used in this study).

Each reconstruction includes an ensemble of 100 members, which reflects the uncertainty associated with each reconstruction (Fig. 2a/b). We use the reconstruction ensemble mean to characterize the 1940s event (Fig. 1) and evaluate the drivers of the event (Figs. 3, 4). For evaluating the rarity of the event, we compare the amplitude of the 1940s event to thousands of years of simulated internal climate variability; this result is sensitive to the precise amplitude of the 1940s anomalies, so we use all ensemble members in the reconstruction to account for uncertainty in this calculation. The reconstruction ensemble reflects different random draws from the climate model priors, so the members only differ by their means (Fig. 2a/b). The temporal variance is the same in each member because the temporal variance is derived from the proxy data, and each member contains information from the same proxy database. We generate an ensemble of independent members by removing the assumption that each datapoint is

autocorrelated in time; we refer to this as "scrambling" the ensemble members (Fig. 2c/d). This has two purposes: (1) each scrambled member becomes a unique realization, allowing us to conduct a rigorous test for uncertainty, and (2) the variance of each scrambled member becomes larger (even though the ensemble variance is the same, as shown in Figs. 2a-d). The larger variance in the scrambled members allows us to compare the reconstructed members to individual members in climate model simulations (more details in the following section), as the variances between these data products is now similar (Fig. 2e/f). More details on the scrambling methods and reconstruction dataset processing can be found in Section 4.1 and Appendix A.

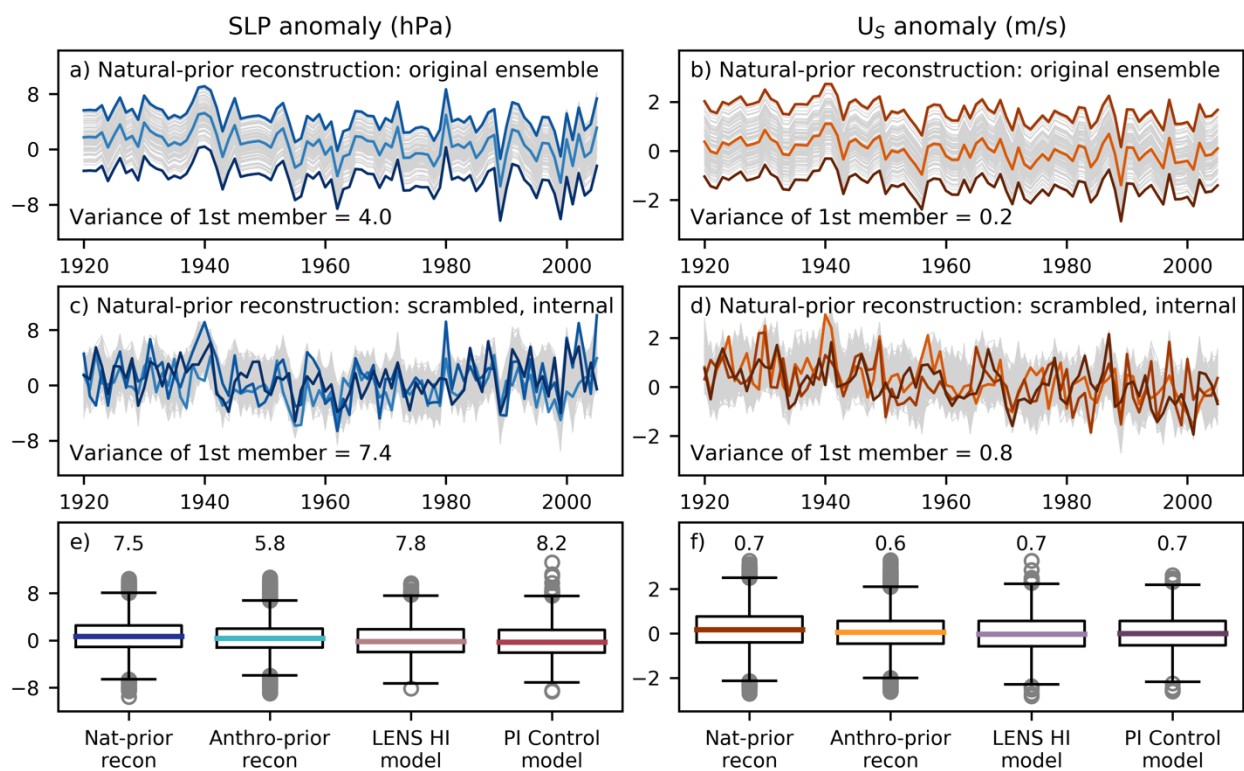

**Figure 2.** Reconstructed and simulated anomalies in SLP (left panels) and $U_S$ (right panels). (a, b) The original 100 ensemble members in the natural-prior reconstruction. (c, d) The 300 ensemble members in the natural-prior reconstruction ensemble, after scrambling the members and isolating the internal component. All timeseries are in anomalies relative to 1961 to 1990. Only three members in each ensemble are plotted in color to highlight the variability of each member (the members containing the minimum, maximum, and median value during 1940); all other members are plotted in gray. (e, f) Box and whisker plots of all data used to calculate the rarity of the 1940s event: two reconstructions (using the internal component of the scrambled ensembles) and two climate model simulations (the internal component of the LENS historical ensemble (LENS HI) and the Preindustrial Control simulation (PI Control)). The colored lines show the medians, the boxes show the inner quartiles, and gray circles show outliers. The variances of each dataset are shown above each boxplot. The corresponding time series of all datasets are shown in Figure A4.

To investigate the origin of the 1940s event, we generate new single-proxy reconstructions, in which we assimilate only ice cores or only coral records. These provide an important measure of the role of local (Antarctic) *vs.* remote

(tropical) proxies in influencing the results. We follow the methods developed by Hakim et al. (2016) and adapted by
O'Connor et al. (2021a) to generate these reconstructions.

**2.2 Climate model simulations**

In addition to the analyses using the single-proxy reconstructions, we investigate sources of variability associated with
the 1940s event by analyzing three historical "pacemaker" ensembles of simulations. These CESM1-based simulations
include historical external forcings and are constrained by instrumental sea surface temperatures from the tropical
Pacific, Indian Ocean Dipole, and North Atlantic (Schneider and Deser, 2018; Yang et al., 2020). The ensemble means
of these simulations reflect the mean response to climate variability in the respective restoring ocean basin and external
forcing.

We evaluate the rarity of the 1940s pressure and wind event by comparing the 1940s anomalies (as characterized by
the scrambled reconstruction ensembles) to thousands of years of simulated internal climate variability. Specifically,
we count the occurrences of similar events in climate model simulations to calculate the frequency of 1940s-like
events in 10kyr of natural climate variability. We use two sets of climate model simulations that reflect pre-industrial
internal climate variability to conduct this comparison: the CESM1 preindustrial control simulation (PI Control) and
the internal component of the CESM1 Large Ensemble Historical ensemble (LENS HI; "I" denotes that the influence
of historical external forcing is removed and we are only considering internal variability). PI Control and LENS HI
are both fully coupled and have ~1° horizontal resolution (Kay et al., 2015). The PI Control is a single simulation of
1,801 years total and contains no anthropogenic forcing. The LENS historical ensemble contains 40 simulations from
1920-2005 for a total of 3,440 years and follows historical anthropogenic forcings over the 20[th] century. We remove
the externally forced component in the historical ensemble to ensure that both simulations reflect internal climate
variability only (timeseries shown in Figure A4; more details in Appendix B). We use these two sets of simulations
because they have low bias in atmospheric circulation in the Amundsen Sea region relative to other climate models
(Holland et al., 2019). Furthermore, the reconstructions lie within the range of states generated by LENS historical
simulations (Holland et al., 2022). Additional details on the methods used for the frequency calculation can be found
in section 4.1.

The LENS simulations allow us to quantify the rarity of the 1940s event relative to 10kyr of internal pre-industrial
climate variability. We note that our calculations are not equivalent to the significance of the event relative to the
Holocene, which experienced differences in variability relating to changes such as insolation and freshwater inputs.
Available transient Holocene simulations are insufficient for conducting this calculation as they are only available at
much lower temporal resolutions and spatial resolutions (e.g., the widely used Transient Climate Evolution of the past
21 ka (TraCE-21 ka) simulations are available only at 3.75° x 3.75° spatial resolution; He et al., 2013). The rarity
calculations presented here are an imperfect analogy to the Holocene but provide a novel estimate of the significance
of the 1940s event, based on the best available simulations.

## 3 Characteristics and Drivers of the 1940s event

### 3.1 Event characteristics

We use the natural-prior reconstruction and the anthro-prior reconstruction ensemble means to investigate the characteristics of the anomalies around 1940 (Fig. 1). Both reconstructions show high pressure and westerly anomalies in the ASE shelf break region lasting approximately five years, from ~1938 to 1942. This demonstrates high confidence that a persistent atmospheric event occurred in the ASE at this time, consistent with previous work

(Schneider and Steig, 2008; Steig et al., 2012; 2013). The spatial patterns of pressure and zonal winds reveal that the shelf-break anomalies are part of an anti-cyclonic feature centered over the South Pacific, just north of the ASE shelf break, emerging as early as 1937. This coincides with local warming patterns and is accompanied by easterly anomalies in the mid-latitude Pacific (Fig. 3).

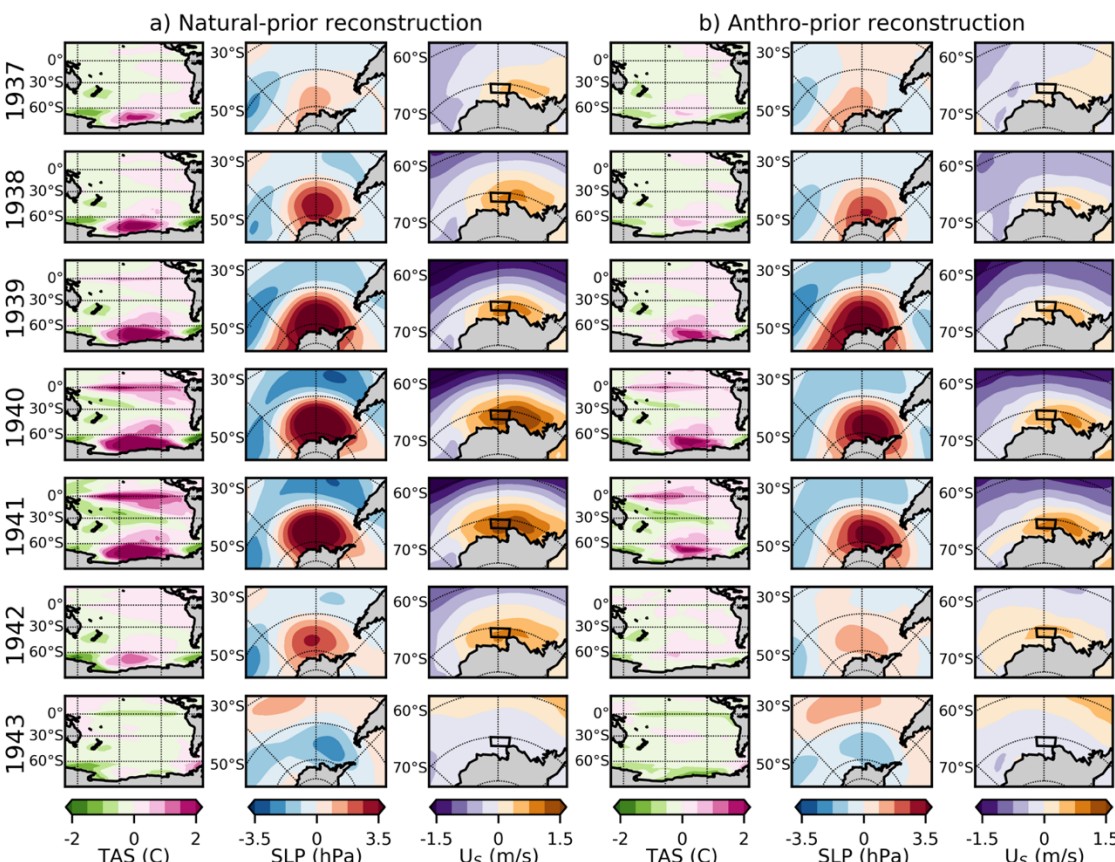

**Figure 3.** (a) Reconstructed anomalies in surface air temperature (TAS), SLP, and $U_S$ in the natural-prior reconstruction (with all proxies assimilated) from the years 1937 to 1943 (the reconstruction ensemble means are shown). (b) Same as in (a) but with the anthro-prior reconstruction. Anomaly reference period is 1961-1990. The black box on the wind panel shows the Amundsen Sea Embayment continental shelf break region.

Both reconstructions agree well with each other and show similar trends toward lower pressure over the 20th century (Fig. 1a). The SLP anomaly reaches a distinct 3-year peak from 1939 to 1941, with the greatest anomaly occurring in 1940. The natural-prior reconstruction maintains a mean anomaly of 4.9 hPa during the three peak years, and the anthro-prior reconstruction reaches a slightly lower peak of 4.3 hPa (anomalies in this section are relative to the period 1979 to 2005 for comparison to ERA5, as shown in Fig. 1). While a similar magnitude anomaly occurred in 1980, it is shorter lived and is not accompanied by a strong westerly anomaly. The 1940s SLP event is the only event in the 20th century that maintains such a large magnitude for three consecutive years (this is also true for 2-year, 4-year, or 5-year averages centered around 1940). Even if the trends in the time series are removed, the 1940 multi-year event remains an outlier relative to the rest of the 20th century. Thus, there is robust evidence from both reconstructions that the SLP anomalies that occurred around 1940 are exceptional relative to the 20th century in terms of combined magnitude and persistence.

Both reconstructions show westerly wind anomalies for at least four years from 1938 to 1942, with a distinct 2-year peak in 1940 and 1941 (Fig. 1b) – one year later and shorter than the distinct peak in SLP. The spatial pattern of SLP indicates that the westerly anomaly reaches its maximum later due to the more southward position of the high-pressure center in 1939 (Fig. 3). The natural-prior reconstruction shows a trend toward easterly conditions over the 20th century, while the anthro-prior reconstruction shows no trend (O'Connor et al., 2021a; Fig. 1b), making the 1940 to 1941 wind anomaly weaker in the anthro-prior reconstruction. The natural-prior wind reconstruction maintains a 2-year peak anomaly of 1.3 m/s from 1940 to 1942, and the anthro-prior reconstruction reaches a weaker peak of 0.5 m/s (relative to an anomaly reference period of 1979-2005). As for SLP, the 1940s wind event is the only one during the 20th century to reach such high magnitudes for two consecutive years. This statement remains true for both reconstructions if we define the wind event using its 3-year, 4-year, or 5-year magnitudes and if we remove the easterly trend. When we use geostrophic winds rather than the directly reconstructed winds, we find that the event has the same timing but greater magnitudes (Fig. 1b), and that it is still a unique event in the 20th century. In short, although the event magnitude is sensitive to the choice of reconstruction, there is robust evidence that a notable multi-year westerly anomaly occurred in the ASE shelf break region centered in 1940-41.

### 3.2 Drivers of the 1940s event

Comparison of the reconstructed ASE SLP with the Nino3.4 Index over the 20th century suggests that much of the 20th century variability is associated with tropical Pacific SST variability (Fig. 1c; r = 0.29 and 0.23 in the natural-prior and anthro-prior reconstructions, respectively; p-values <0.05), as earlier studies have established (e.g., Lachlan-Cope & Connolley, 2006; Ding et al., 2011; Steig et al., 2012; Holland et al., 2019). Previous work has suggested that the 1940s event in the Amundsen Sea was a response to the large ~1940-42 El Niño event (Schneider and Steig, 2008; Steig et al., 2012) which was unusually persistent (Bronnimann et al., 2004). Our reconstructions are consistent with that suggestion, as they show high pressure and westerly anomalies characteristic of a classic Rossby wave train response during El Niño years. However, the reconstructions also show that the South Pacific warming and high-

pressure anomalies led the onset of the 1940 El Niño by up to two years (Figs. 1c, 3), suggesting that they are not exclusively a response to tropical Pacific convection. Instead, the high pressures and shelf-break westerlies from 1938 to 1942 may be a compounded response to both tropical Pacific convection and local internal variability or teleconnections with regions other than the tropical Pacific (Li et al., 2021).

To further investigate the origin of the anomalies in the Amundsen Sea, we conduct single-proxy experiments. We generate two new types of reconstructions: one in which we assimilate only ice core records, and one in which we assimilate only coral records. We use the same proxy database, ensemble Kalman filter method, and two priors (one with natural forcing and one with anthropogenic forcing) as in O'Connor et al. (2021a). The ice-only reconstruction reveals signals captured in Antarctica (primarily West Antarctica, which has the greatest number of high-resolution ice-core records); the coral-only reconstruction highlights signals captured in the tropics (primarily the tropical Pacific). For simplicity, we present the results from the natural-prior single-proxy experiments in the main text (using the reconstruction ensemble mean), but our results are similar when we use the anthro-prior (Fig. A2). The ice-only reconstruction shows large warming, high pressure, and westerly anomalies in the Amundsen Sea starting as early as 1938 (Fig. 4). In the coral-only reconstruction, those signals do not emerge until 1940 and 1941, further indicating that the high pressure and large westerly anomalies detected in the Amundsen Sea cannot exclusively be a response to tropical Pacific variability (Figs. 1c, 4).


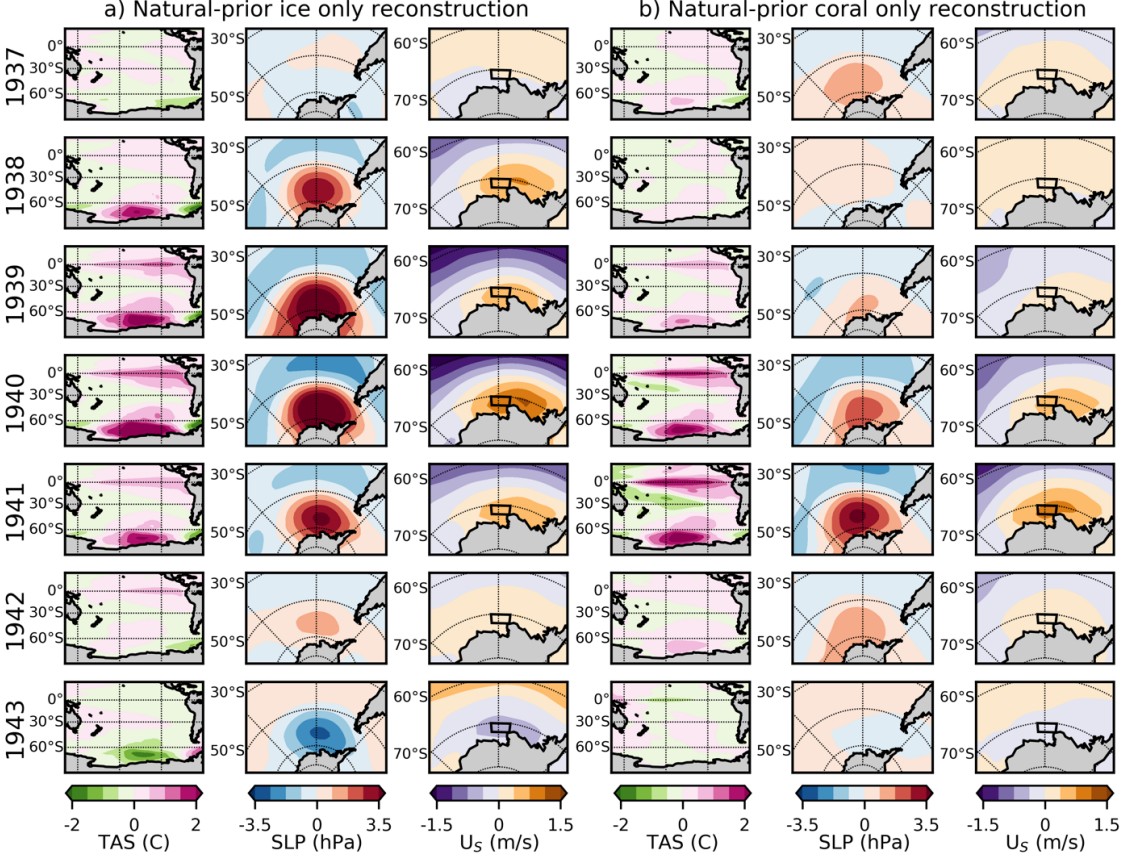

**Figure 4.** Same as in Figure 3, but with the natural-prior and only ice cores assimilated (a) and only corals assimilated (b). The same figure using the anthro-prior is shown in Figure A2.


It is possible, in principle, that the differences in timing of local anomalies and tropical Pacific anomalies could reflect dating uncertainty in the proxy records. However, this is unlikely: the West Antarctic ice cores have a demonstrated dating uncertainty of less than 1 year owing to the use of multiple volcanic markers of known age, and unambiguous seasonal variations in chemistry (Steig et al., 2005). Furthermore, even higher resolutions are available in tropical

Pacific coral records, enabling dating uncertainties of one to two months in modern corals and less than 1 year in fossil corals (Cobb, 2002; Sanchez et al., 2020; O'Connor et al., 2021b). As an additional test that does not depend on proxy data, we analyze the tropical Pacific pacemaker ensemble v1– a set of 20 simulations constrained to follow tropical Pacific sea surface temperatures (SSTs) starting in 1920 (the same simulations used to form the prior in the anthro-prior reconstruction). The ensemble mean of the Pacific pacemaker simulations – which represents the response to

tropical Pacific variability and external forcing – shows high pressures and shelf break westerlies in the ASE only in 1940 and 1941 (Fig. 5), consistent with the results from our single-proxy experiments. The individual members in the Pacific pacemaker ensemble represent realizations of climate variability driven by influences outside of the tropical Pacific; several members show high pressure anomalies in the Amundsen Sea and shelf break westerlies in 1938 and 1939, demonstrating that these patterns can indeed emerge even in the absence of tropical Pacific forcing (example

members shown in Fig. 5). These simulations bolster the results from the single-proxy reconstructions: it is unlikely that the 1940s event over the Amundsen Sea was exclusively a response to the 1940-42 El Niño.

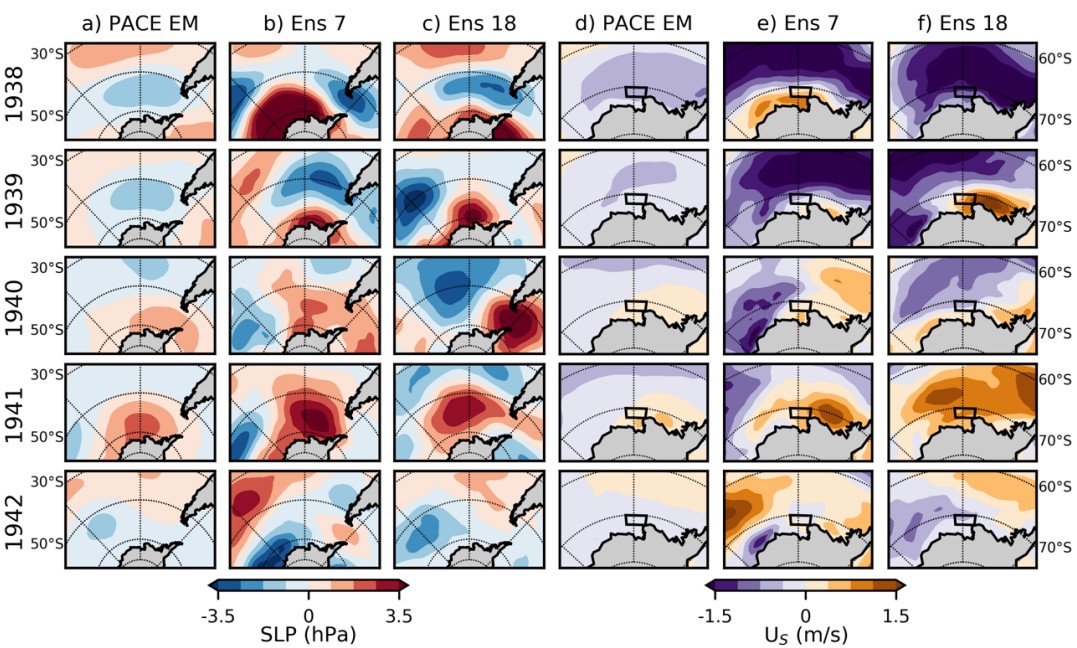

**Figure 5.** (a-c) Modeled SLP and (d-f) $U_S$ anomalies from 1938 to 1942 in the tropical Pacific pacemaker ensemble of simulations. The ensemble mean, ensemble 7, and ensemble 18 of the Pacific pacemaker ensemble is shown. Anomaly reference period is 1961 to 1990.

To investigate additional sources of variability associated with the event in the ASE, we next evaluate two additional
CESM1 pacemaker simulations: one constrained to follow observed SSTs from the North Atlantic, and one constrained to follow observed SSTS from the Indian Ocean (Yang et al., 2020). Both simulations include historical external forcing. Like the tropical Pacific pacemaker simulation, the ensemble means of these simulations reflect the mean response to observed variability in the respective restoring SST ocean basin and external forcing. The SLP and $U_S$ anomalies in all three pacemaker simulations are shown in Fig. 6. In 1939, high pressure anomalies are found in
the Indian and Atlantic simulations; in 1940, high pressures are shown in the Pacific and Atlantic simulations; and in 1941, high pressures are shown in the Pacific and Atlantic simulations. The results for $U_S$ are more subtle: westerly anomalies are shown in the Pacific simulation in 1940 and 41, as previously mentioned, and weakly westerly anomalies are shown in the Atlantic pacemaker simulation from 1939 to 1941. These results suggest that the large anticyclonic anomalies shown in the reconstructions may be a result of a confluence of different climate modes
operating in succession.

The results from the Atlantic and Indian pacemaker simulations may be considered reliable only if they are broadly consistent with tropical Pacific variability, which is indeed the case for the Indian pacemaker simulations in 1938 and 1939, and for the Atlantic pacemaker simulation from 1939 to 1941 (TAS anomalies in all three pacemaker simulations
are shown in Fig. A3). However, we note that it is difficult to isolate the variability between the Pacific and other ocean basins (i.e., the observed North Atlantic warming may be associated with the El Niño). Thus, while the results from these simulations are associated with large uncertainties, they provide insight into the potential sources of other variability that may explain the 1940s event, consistent with previous work (e.g., Okumura et al., 2012; Li et al., 2021). Combining the results from the single-proxy experiments, the tropical Pacific pacemaker simulation, and these two
additional pacemaker simulations, the evidence suggests that the 1940s anti-cyclonic anomalies over the Amundsen Sea are associated with a combination of factors resulting in a potentially rare "perfect storm".

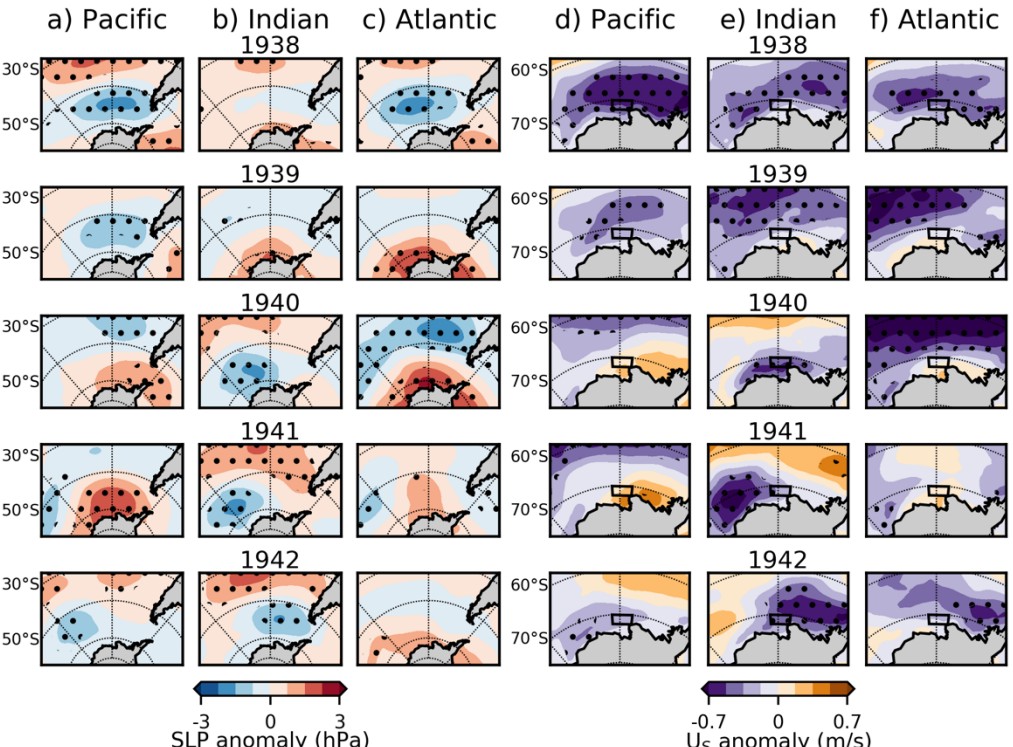

**Figure 6.** Modeled SLP anomalies from 1938 to 1942 in the ensemble mean of the (a) tropical Pacific, (b) Indian Ocean, and (c) North Atlantic pacemaker simulations. Stippling is shown where anomalies are greater than 1 standard deviation from the mean (no regions contain anomalies greater than 2 standard deviations). (d-f) Same as in a-c, but for $U_S$ anomalies. Anomaly reference period is 1961-1990. We note that the color bar in this figure is smaller than that of Figure 5 due to the smaller variability in the ensemble means of the simulations (Figure 5 shows the ensemble mean and individual members from the Pacific pacemaker simulations).

**4 Rarity of the 1940s event in a natural climate**

The 1940s event in the ASE is notable because it is likely a combined response to both an exceptionally persistent El Niño event, and to other sources of variability occurring in preceding years. This suggests that the frequency of this type of event in the ASE is not comparable to the frequency of El Niño events; it is probably much rarer. Quantifying the frequency of the 1940s ASE pressure and westerly anomalies in the context of internal climate variability is crucial for investigating the question of why glacier retreat may have started in the 1940s, as suggested by the sediment-core evidence (Smith et al., 2017). Was the ongoing ice retreat triggered solely by a very unlikely natural event in the 1940s (e.g., an event that is unprecedented in millennia)? Or was that event a relatively common occurrence that only triggered ice retreat in 1940 in combination with other conditions favorable for retreat? In this section, we calculate the frequency of the 1940s anomalies occurring in a natural climate by looking for similar occurrences within thousands of years of climate model simulations without anthropogenic forcing. We evaluate the statistics of these occurrences to investigate whether the 1940s event may be exceptional in the context of 10kyr of climate variability.

## 4.1 Frequency calculation

To estimate the frequency of the 1940s event, we use reconstructions to quantify the magnitude of the event and then search simulations of internal climate variability for events that meet or exceed that magnitude. We explain the details of the calculations here; the results for SLP and $U_S$ and reviewed in sections 4.2 and 4.3, respectively.

To quantify the magnitude of the 1940s event, we use both the natural-prior and anthro-prior reconstructions. As 390 described in section 2.1, we use the scrambled reconstructions ensembles to ensure that uncertainty is fully accounted for and that the variances of all datasets used in the calculation are comparable (Figs. 2e/f). As described in section 2.2, we compare the reconstructed magnitudes to internal climate variability as captured by the PI Control (a single simulation without anthropogenic forcing) and LENS HI (an ensemble of historical simulations with the externally forced component – quantified as the ensemble mean of the LENS historical ensemble – removed). We process all 395 datasets to ensure that they all reflect the internal component of climate variability and contain the same anomaly reference periods (more details in Appendix B; timeseries are shown in Fig. A4).

With the two reconstruction ensembles and two sets of climate model simulations, we calculate the frequency of the 1940s event in a natural climate. To go through an example of the calculation, let us use the natural-prior SLP 400 reconstruction (using the internal component of the scrambled ensemble) to quantify the 1940s event magnitude, which we define as the 5-year mean centered on 1940 (1938 to 1942). We use the ensemble to quantify the mean magnitude and its 95% confidence interval, as shown in Figure 7a (see the values for the 5-year window length to follow this example). Next, we use the PI Control simulation to evaluate the statistics of the 5-year magnitude relative to internal climate variability. We calculate the means of all 5-year windows in the PI Control simulation of SLP, 405 sampling with a 50% overlapping window. As a simple statistic for evaluating the statistical significance of the reconstructed 5-year magnitude relative to internal variability, we calculate the sigma level of the magnitude, i.e. the number of standard deviations from the simulated 5-year samples (assuming Gaussian statistics; Fig. 7c). Next, we calculate the frequency of the 5-year magnitude as follows: if any 5-year SLP sample from the simulation has a mean magnitude at least as great as the reconstructed 5-year magnitude, it counts as an occurrence. The total number of 410 occurrences divided by the sample size (the number of 5-year windows in the simulation) equals the fractional probability; we multiply this probability by 10,000 to yield the frequency of the 1940s event per 10 kyr (Fig. 7c). The calculations are sensitive to the precise magnitude used to characterize the 1940s event, so we propagate the confidence intervals from the reconstructed magnitudes (Fig. 7a) throughout the sigma and frequency calculations (i.e., we repeat the calculation using the upper and lower bounds of the reconstructed magnitude; results are shown as 415 error bars in Figs. 7c, 7e).

We use the sigma level to quantify the statistical significance of the event. Because a statistically significant event does not necessarily explain the start of glacier retreat, we use the frequency estimates to evaluate the narrative that the 1940s event may explain the start of glacier retreat in West Antarctica. If the 95% confidence interval of the

frequency includes 1 per 10kyr, we fail to reject the hypothesis that the event is unprecedented. If the confidence interval includes <20 events per 10kyr , we classify the event as "exceptional". If the event fails to meet these criteria but is a 2-sigma event, we classify it as "relatively uncommon". We note that the number 20 for "exceptional" is relatively arbitrary; it is simply used to define a threshold for qualitative descriptions of the results, and our main conclusions are insensitive to this precise choice (i.e., the conclusions are unchanged if we choose 5 or 50).

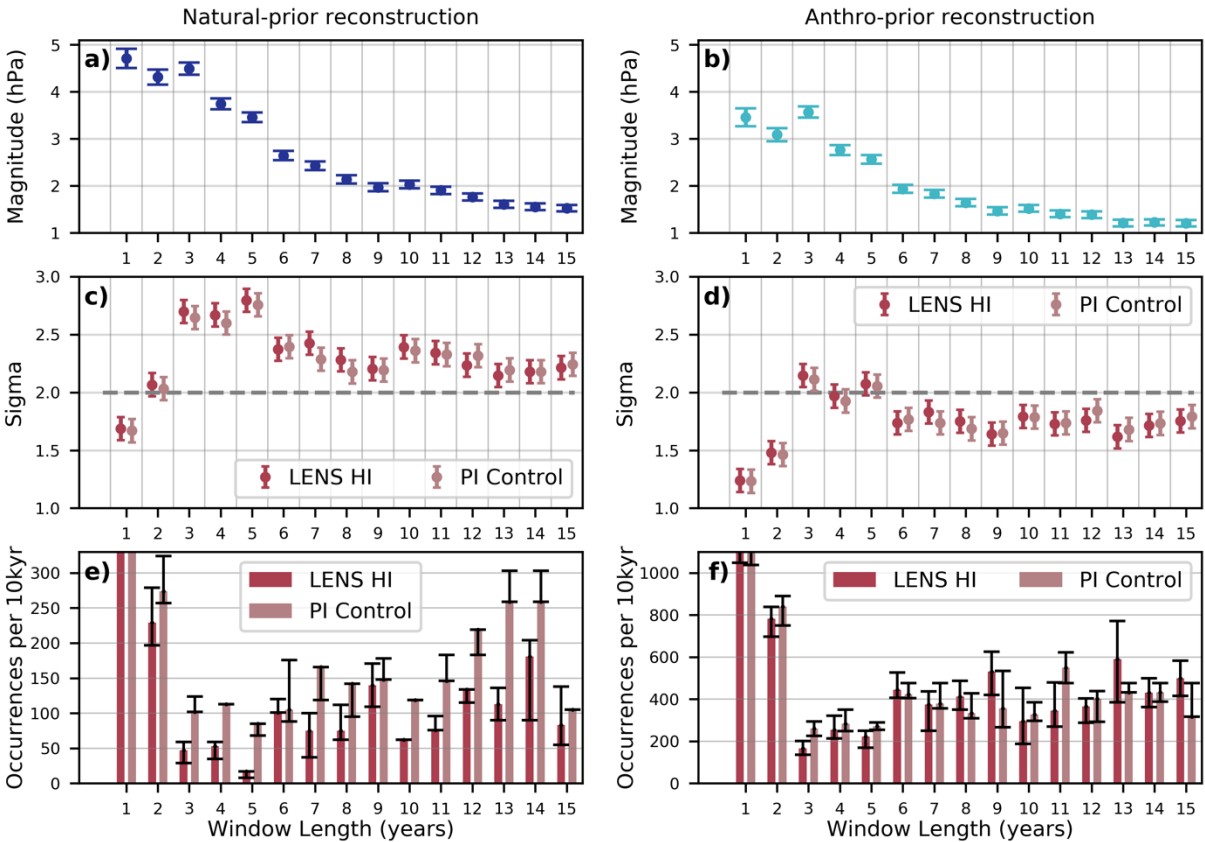


**Figure 7.** Magnitudes of the 1940s SLP event based on 1-to-15-year averaging windows centered around 1940/41, calculated using (a) the natural-prior reconstruction and the (b) anthro-prior reconstruction (in anomalies relative to 1961 to 1990). The circles represent the mean magnitudes and the error bars represent the 95% confidence intervals. (c, d) Sigma levels of each reconstructed magnitude in panels (a) and (b) relative to two climate model simulations: 430 the Preindustrial Control simulation (PI Control) and the internal component of the LENS Historical simulation (LENS HI). The dashed line highlights the 2-sigma level. (e, f) Histograms of the frequency of the 1940s event based on each averaging window, reported as occurrences per 10kyr (using the magnitudes in (a) and (b)) found in the two climate simulations. The error bars represent occurrences based on the upper and lower bounds of the confidence intervals in (a). The y-axes are selected to highlight the rarest events and differ between (e) and (f); the values for all events can 435 be found in Table 1.

We repeat the magnitude, frequency, and sigma calculations using both reconstructions, corresponding to the left and right columns in Figure 7. We also repeat the calculation using both sets of climate simulations, corresponding to the two colors in Figures 7c-f. We use averaging windows between one and fifteen years long centered around the year 440 1940, allowing us to test the significance of the event in terms of both amplitude and duration (for even numbers, we select 1940.5 as the center value to sit between the different peaks of the event in SLP and Us). These averaging windows correspond to the x-axis on Figure 7. We note that our overall conclusions are insensitive to the precise

choice of center date; for example, they remain unchanged if we instead use 1942 as the end date of the averaging window (which excludes the more negative values in the mid 1940s and includes the more positive values around
1930). Results from the same set of calculations for $U_S$ are shown in Figure 8.

## 4.2 Frequency of the 1940s pressure event

Using the natural-prior reconstruction, the magnitudes and 95% confidence intervals for the 1940s SLP event based
on each *x*-year window are shown in Figure 7a. As expected, the 1 to 3-year windows show the greatest magnitudes, and longer windows show gradually decreasing magnitudes. While the 1-year window shows a similar magnitude to the 2 and 3-year windows, it does not exceed the 2-sigma level relative to internal climate variability; the 2 to 15-year window magnitudes all exceed the 2-sigma level, providing evidence that the 1940s SLP event is indeed significant relative to internal climate variability in terms of its combined magnitude and persistence (Fig. 7c). For the frequency
calculation, the absolute number of event occurrences is sensitive to the choice of climate model simulation (LENS HI vs. PI Control), but both simulations show that the 5-year window magnitudes yield the fewest occurrences (Fig. 7e, Table 1). Based on the 5-year window magnitudes, we estimate 17 and 85 occurrences per 10kyr in the LENS HI simulation and PI control simulation, respectively, with lower confidence interval bounds of 8 and 68, respectively. None of the results yields confidence intervals that include 1 or fewer occurrences, so there is evidence (based on this
reconstruction) to reject the hypothesis that the event is unprecedented. Based on the classification criteria outlined above, these results provide evidence that, using the 5-year mean (the rarest mean) from the natural-prior reconstruction, the 1940s SLP event is relatively uncommon (i.e., it is a 2-sigma event with a mean estimate of 20-90 occurrences per 10kyr). The SLP event can be classified as relatively uncommon based on its mean using all windows from 3 to 15 years long, suggesting that the event is notable on interannual to interdecadal timescales.


**Table 1.** Number event occurrences per 10kyr found in the PI Control and LENS HI climate model simulations, based on mean event magnitudes from the natural-prior reconstruction and the anthro-prior reconstruction. The lower bound of occurrences per 10kyr (based on the 95% confidence interval of event magnitudes) is shown in parentheses. The minimum values in each column are bolded. The numbers correspond to the histograms in Figures 7 and 8.


| | SLP | | | | Us | | | |
|---|---|---|---|---|---|---|---|---|
| | Natural-prior reconstruction | | Anthro-prior reconstruction | | Natural-prior reconstruction | | Anthro-prior reconstruction | |
| Years in event | LENS HI | PI Control | LENS HI | PI Control | LENS HI | PI Control | LENS HI | PI Control |
| 1 | 529 (465) | 544 (455) | 1200 (1049) | 1149 (1038) | 476 (392) | 555 (427) | 1162 (988) | 1193 (999) |
| 2 | 229 (197) | 274 (257) | 782 (697) | 840 (750) | 76 (58) | 72 (56) | 370 (300) | 352 (302) |
| 3 | 47 (29) | 102 (102) | **166 (136)** | **260 (226)** | 53 (47) | 68 (45) | 375 (297) | 487 (374) |
| 4 | 53 (35) | 113 (113) | 255 (214) | 283 (249) | 53 (35) | 22 (11) | **250 (184)** | **260 (181)** |
| 5 | **17 (8)** | **85 (68)** | 223 (169) | 272 (255) | 35 (8) | **17 (17)** | 392 (303) | 391 (323) |
| 6 | 101 (101) | 105 (88) | 444 (407) | 423 (405) | 37 (27) | 35 (35) | 592 (518) | 511 (423) |
| 7 | 75 (37) | 166 (119) | 375 (250) | 380 (357) | 75 (25) | 23 (23) | 625 (487) | 666 (595) |

| | | | | | | | | |
|---|---|---|---|---|---|---|---|---|
| 8 | 75 (62) | 142 (95) | 412 (350) | 333 (309) | 62 (37) | 47 (47) | 600 (437) | 666 (595) |
| 9 | 140 (109) | 148 (148) | 531 (421) | 357 (267) | 31 (15) | 29 (29) | 640 (546) | 684 (595) |
| 10 | 62 (62) | 119 (119) | 296 (187) | 327 (297) | **0** **(0)** | 29 (29) | 406 (359) | 505 (386) |
| 11 | 76 (76) | 146 (146) | 346 (269) | 549 (476) | 19 (19) | 36 (36) | 538 (403) | 549 (476) |
| 12 | 134 (115) | 219 (183) | 365 (288) | 402 (293) | **0** **(0)** | 36 (0) | 365 (269) | 402 (366) |
| 13 | 113 (90) | 259 (259) | 590 (386) | 432 (432) | **0** **(0)** | 43 (43) | 454 (431) | 606 (476) |
| 14 | 181 (90) | 259 (259) | 431 (363) | 432 (389) | 68 (45) | 86 (86) | 500 (477) | 649 (606) |
| 15 | 83 (55) | 105 (105) | 500 (416) | 317 (317) | 27 (27) | 52 (52) | 666 (611) | 476 (423) |

When we repeat the experiment using the anthro-prior reconstruction, the significance weakens substantially due to the lower SLP amplitudes (Fig. 7b). In this case, the 3 to 5-year magnitudes are again the most notable, but only the 3-year mean exceeds the 2-sigma level (Fig. 7d). The 3-year magnitudes are the rarest (Fig. 7f), with 166 and 260 occurrences per 10kyr using LENS HI and the PI Control, respectively (the lower bounds are 136 and 226, respectively). We note that the anthro-prior SLP reconstruction members have a smaller variance than the simulations and the natural-prior reconstruction members (Fig. 2e), which may partially explain the higher frequency estimates found using the anthro-prior reconstruction. Like the results from the natural-prior reconstruction, none of the confidence intervals includes 1 or fewer occurrences per 10kyr, so there is evidence from both reconstructions to reject the hypothesis that the SLP event is unprecedented. Based on the 3-year mean (the rarest mean) in the anthro-prior reconstruction, we again classify the 1940s SLP event as relatively uncommon (i.e., it is a 2-sigma event with a mean estimate of ~200 occurrences per 10kyr).

Together, the results from both reconstructions show that the 1940s SLP event is most notable in terms of its 3 and 5-year means, and that the event is relatively uncommon compared to 10kyr of internal climate variability. It is unlikely that the 1940s SLP event is unprecedented in 10kyr of internal pre-industrial climate variability; the results show a mean estimate of ~20 to 200 occurrences per 10kyr.

**4.3 Frequency of the 1940s zonal wind event**

For $U_S$, we find broadly similar results but with greater sensitivity to the choice of reconstruction, as expected from our earlier analyses. Using the natural-prior reconstruction, we find that the 1 and 2-year windows have the greatest magnitudes. After the 2-year mark, the magnitudes decline as the window size increases (Fig. 8a), and all 2 to 15-year magnitudes exceed the 2-sigma level relative to internal climate variability (Fig. 8c), demonstrating that the 1940s $U_S$ event is statistically significant on interannual to interdecadal timescales. Using the LENS HI simulation, the 10-year magnitude yields the rarest frequencies of 0 occurrences per 10 kyr (the 12-year and 13-year means also yield 0). Using the PI Control simulation, the 5-year mean yields the rarest frequencies of 17 per 10kyr (Table 1; the lower bound is also 17). Using the 12-year mean, both simulations show lower bounds of 0 (Table 1); thus, based on the

natural-prior reconstruction 12-year mean, we fail to reject the hypothesis that the 1940s $U_S$ event may be unprecedented (with an expected ~0 to 30 occurrences per 10kyr). Based on the 5-year mean, the confidence intervals include <20 occurrences per 10kyr (with an expected ~20-30 occurrences per 10kyr), so the 1940s $U_S$ event can be classified as exceptional.

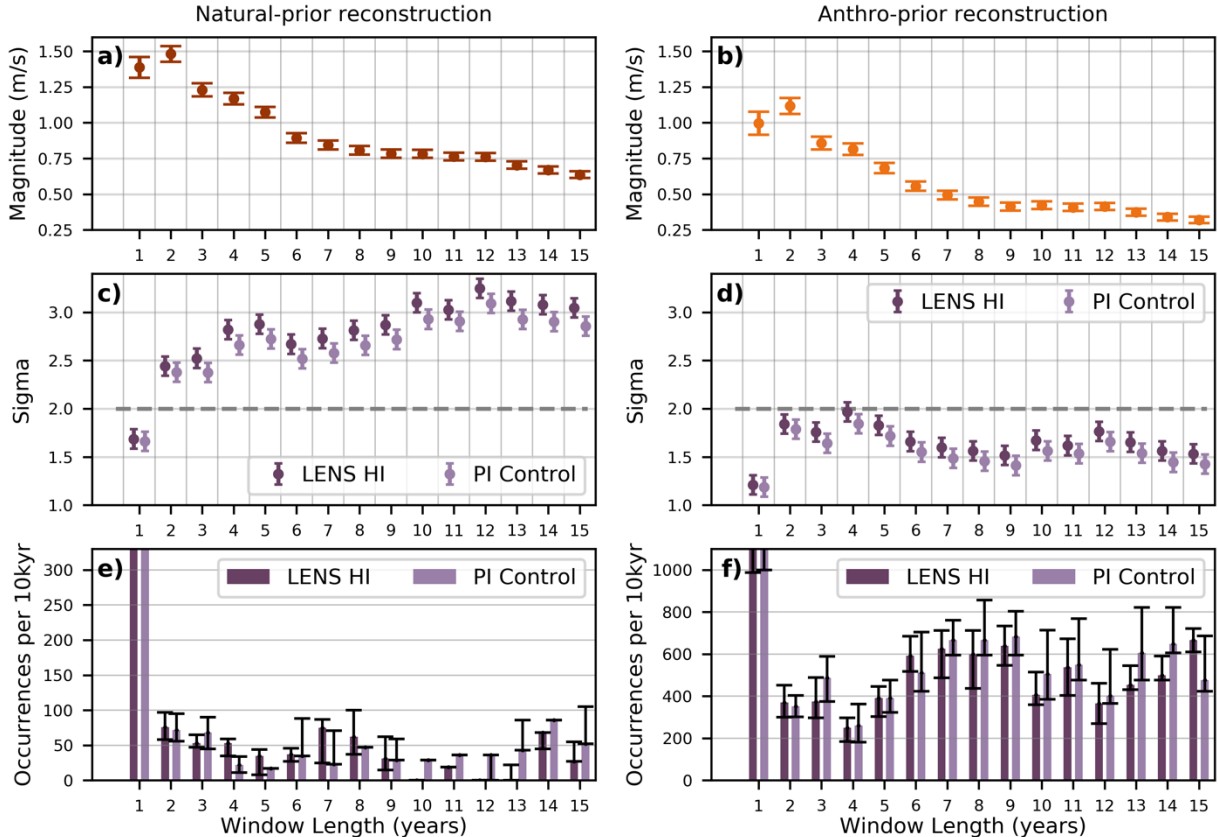

**Figure 8.** Same as in Figure 7, but for reconstructions of $U_S$ (geostrophic winds are used for the anthro-prior reconstruction). The values for all events can be found in Table 1.

We repeat this analysis with the anthro-prior geostrophic wind reconstruction, as it has greater skill (*vs*. ERA5) than the directly reconstructed winds (our main conclusions are insensitive to this choice). Like SLP, we find weaker event magnitudes in the anthro-prior reconstruction compared to the natural-prior reconstruction (Fig. 8b), lowering the calculated significance of the events. The 4 to 5-year magnitudes are the most exceptional, but none of the magnitudes exceeds the 2-sigma threshold (Fig. 8d). The weaker magnitudes in this reconstruction yield greater event occurrences (Fig. 8f, Table 1). The 1-year magnitude is the most common; the 4-year magnitude is the rarest, with 250 and 260 occurrences per 10kyr in the LENS HI and LENS PI simulations, respectively (and lower bounds of 184 and 181, respectively). Again, there is local minimum at the 12-year mark, which yields 365 and 402 occurrences per 10kyr in the LENS HI and PI Control simulations, respectively. The results from the anthro-prior reconstruction provide insufficient evidence that the 1940s $U_S$ event may be considered unprecedented, exceptional, or relatively uncommon; they suggest that the event is relatively common (the minimum frequency found is ~180 occurrences per 10kyr).

To summarize the results for $U_S$, using the natural-prior reconstruction, we find evidence that the 1940s $U_S$ event could

be unprecedented relative to 10kyr of pre-industrial climate variability (based on the 12-year mean) or exceptional (based on the 5-year mean). However, the results from the anthro-prior wind reconstruction do not support this; they instead suggest that the 1940s westerly event is relatively common and not statistically significant by any metric. The results from both reconstructions show that the 1940s $U_S$ event is most notable for its combined long duration and large magnitude, especially based on ~5-year and ~12-year means (with expected occurrences of ~0-400 per 10kyr).

We note that the 12-year magnitude comprises approximately half of the amplitude of the shorter window magnitudes (in both the natural-prior and anthro-prior reconstructions), suggesting that the 1940s $U_S$ anomalies may be a result of both interannual variability (i.e., the El Niño event) and interdecadal variability. The calculations for $U_S$ are not hindered by the same uncertainties associated with the SLP analysis: the variances among all datasets are similar (Fig. 2f), and the probabilities are generally insensitive to the choice of simulation used. Therefore, the differences in the

probabilities from the natural-prior reconstruction and the anthro-prior reconstruction are a result of the uncertainties in the reconstruction magnitudes (resulting from the different priors used in data assimilation).

To summarize the results for both variables (using both reconstructions and both simulations), we find evidence that the 1940s SLP event can be classified as relatively uncommon (i.e., statistically significant with ~20-200 occurrences

estimated per 10kyr based on 3 to 5-year means). We find insufficient evidence that the 1940s $U_S$ event can be classified as unprecedented, exceptional, or relatively uncommon, due to the lower magnitudes found in the anthro-prior reconstruction (the minimum frequency found using the anthro-prior reconstruction is ~180 occurrences per 10kyr, based on the 4-year mean; the expected frequency based on the 4-year mean in both reconstructions is ~20 to 260).


## 5    Discussion

In this study, we use proxy-based reconstructions of SLP and $U_S$ to characterize the 1940s atmospheric anomalies over the Amundsen Sea, West Antarctica, which have previously been identified as a candidate for initiating glacier

retreat in this region. The reconstructions show high pressure and westerly anomalies over the Amundsen Sea for at least five years, centered around the years 1940 and 1941. We find that the event may be a "perfect storm" of atmospheric circulation associated with the very strong El Niño event and potentially coinciding with Indian Ocean and Atlantic Ocean variability. Our rarity calculations show that the 1940s westerly event (in terms of its 4-year mean) is expected to occur ~20 to 260 times per 10kyr of pre-industrial climate variability, suggesting that the wind event is

likely not exceedingly rare.

The rarity is highly sensitive to the choice of reconstruction used. Our findings emphasize the importance of considering multiple climate model simulations as priors in proxy-based reconstructions generated using data assimilation. We find that the sensitivity to the climate-model prior is especially large for zonal winds, as the difference

in reconstructed 20[th] century wind trends influences the magnitude of the 1940s anomalies (Fig. 1). Anthropogenic

forcing may alter the global teleconnection patterns that dictate the information gained from proxy data in the assimilation process, perhaps making the trends in the anthro-prior reconstruction more realistic. However, the skill in the anthro-prior reconstruction (relative to ERA5) is slightly weaker than that of the natural-prior reconstruction, leaving the uncertainties too large to favor the results from either reconstruction in our conclusions; the true history likely lies between the two reconstructions.

Our finding that the 1940s event in the ASE is evidently a response to multiple sources of variability highlights the limitations of using datasets constrained by tropical Pacific variability alone (e.g., Holland et al., 2019). While the tropical Pacific is the dominant source of interannual variability in the Amundsen Sea region (Lachlan-Cope and Connolley, 2006; Ding et al., 2011; 2012), other sources of variability can have a substantial impact on large-magnitude events in the ASE. This further illustrates the importance of paleoclimate reconstructions around West Antarctica that include global proxy data, consistent with the findings of Holland et al. (2022), who show that variability arising from regions other than the tropical Pacific can have a large impact on trends in the ASE.

This study is the first to quantify the significance of the 1940s event in the ASE, but our results are associated with several uncertainties as mentioned above. Additionally, we note that we only use simulations from a single native climate model (CESM) for our probability analysis. We selected this model because it has the least bias in the ASE region, and the reconstructions lie within the range of simulated states (Holland et al., 2019; 2022), but some biases inevitably remain (e.g., in the precise position of the winds). Furthermore, the internal components of the LENS simulations are an imperfect analogy to the Holocene, which was subject to differences in insolation, freshwater inputs, and possibly El Niño/Southern Oscillation (ENSO) variability (e.g., Mayewski et al., 2004). The results presented here are associated with uncertainty but provide an estimate of the significance of the 1940s event based on the best available simulations.

Another caveat is that our analysis focuses on the shelf-break region. Many previous studies have argued that this is the region most closely associated with the poleward transport of warm CDW (Thoma et al., 2008; Holland et al., 2019; Naughten et al., 2022), and it is close to the spatial peak of the reconstructed 1940s zonal wind anomalies (Fig. 3). However, the relationship between local winds and ocean circulation in this region is complex, as illustrated by regional ocean simulations (e.g., Thoma et al., 2008; Webber et al., 2019; Dotto et al., 2019; Naughten et al., 2022) and oceanographic observations (e.g., Assmann et al., 2013; Wåhlin et al., 2013; Walker et al., 2013; Dutrieux et al., 2014; Kim et al., 2017; Jenkins et al., 2018; Wåhlin et al., 2021). Indeed, recent work suggests that the influence of local shelf-break winds on ocean circulation may change sign on longer timescales (Silvano et al., 2022). Furthermore, large scale atmospheric and oceanic circulation must also influence CDW transport in the ASE (Nakayama et al., 2018), though the mechanisms related to remote sources of variability have not been comprehensively examined.

While our frequency analysis suggests that the 1940s $U_S$ event is relatively common on millennial timescales, it remains likely that this event played an important role in triggering the current stage of glacier retreat, as suggested

by the evidence from sediment cores (Smith et al., 2017). Similar westerly anomalies probably occurred previously during the last 10kyr, perhaps causing short-lived retreats that the ice recovered from once the atmospheric and associated oceanic perturbation ceased. We propose that the 1940s anomaly may have been the first such event to be superimposed on wider climatic, oceanic, and/or glacial conditions favorable for initiating prolonged retreat (e.g., multi-decadal ocean variability or large-scale warming, or a particularly sensitive ice-sheet grounding line position (Christianson et al., 2016; Jenkins et al., 2018; Christian et al., 2021)). Alternatively, the 1940s event may have caused a similar ice perturbation to previous atmospheric anomalies but was unusually followed by conditions suitable for ongoing retreat, preventing the ice from recovering from this particular perturbation (i.e., ice loss is a result of both a historical trigger and more recent changes in wind/ocean forcing). For example, anthropogenically forced wind changes driven by greenhouse gas emissions and ozone depletion emerged shortly afterwards, in the latter half of the 20[th] century (Holland et al., 2022). Alternatively, recent changes in buoyancy, perhaps due to increased freshwater inputs locally or from the Antarctic Peninsula, could have played in a role in preventing ice recovery after the 1940s event (e.g., Webber et al., 2019; Flexas et al., 2022). These hypotheses are consistent with the available evidence but require further investigation.

## 6    Conclusions

In this study, we use paleoclimate reconstructions and climate model simulations to place novel constraints on the 1940s atmospheric event as a candidate for initiating glacier retreat in the ASE. Using two paleoclimate reconstructions that reflect the range of uncertainty in the available proxy-constrained reconstructions from this time, we find that a large anticyclonic anomaly and strong westerlies occurred in this region with a distinct peak from ~1938-1942, consistent with previous work (Schneider and Steig, 2008). The differences between the reconstructions underscore the importance of considering multiple climate-model priors – and considering anthropogenic forcing – when analyzing reconstructions generated using proxy-data assimilation. The results from our single-proxy experiments and the tropical Pacific pacemaker simulations provide evidence that the event was a combined response to an anomalously persistent El Niño event combined with variability not associated with the tropical Pacific.

In comparison to climate model simulations, the 1940s pressure and zonal wind anomalies are most notable for their combined large magnitude and long duration, with the magnitudes maintained for ~5 years and 10 years being the most significant relative to internal climate variability. Our rarity estimates show that the 1940s pressure and zonal wind anomalies are likely to occur tens to hundreds of times in 10kyr or internal climate variability, suggesting that the event may not be particularly exceptional. Our results reveal new uncertainties associated with the narrative that the westerly event may have triggered West Antarctic glacier retreat. The 1940s event may have been the initial atmospheric perturbation, however given the estimated likelihood of this type of event, other factors are a necessary component to the narrative. It is unlikely that the 1940s atmospheric perturbation alone can explain the current stage of glacier retreat in the ASE. We suggest that if the event were superimposed on favorable oceanic or glaciological conditions, or followed by anthropogenically forced trends, the event may have played a role in initiating ice loss.

Ocean simulations forced by realistic climate histories, and continued direct observations in the field, are needed to better constrain the mechanisms responsible for glacier retreat in West Antarctica.

## Appendix A. Scrambling the reconstruction ensemble members


To generate a scrambled member, we construct the timeseries by randomly drawing a value from the original ensemble for each year. This removes the assumption that each datapoint is autocorrelated in time, serving as a rigorous test for uncertainty. We do this timeseries construction 300 times, producing 300 scrambled reconstruction members. The resulting scrambled ensemble has the same ensemble mean and intra-ensemble variance as the original ensemble for

any given year, but the members are now unique realizations (i.e., the gray shaded regions in Fig. 2a and 2c are the same, but the temporal variance in each individual member is different, as annotated on the subpanels). Each scrambled member now has a temporal variance closer to that of the climate model simulations, allowing us to compare the reconstructions to the simulations without major discrepancies in variances (see the variances of all datasets annotated in Fig. 2e/5f). We note that the anthro-prior reconstruction ensemble is slightly smaller in variance

than the climate model simulations for SLP, which adds additional uncertainty to our SLP analysis using the anthro-prior reconstruction.

## Appendix B. Data processing for the rarity calculation

We take several data processing steps to ensure that the scrambled reconstruction ensembles and climate model simulations are directly comparable. The reconstructions and LENS HI are ensembles that contain members from the period 1920 to 2005. The PI Control simulation is one simulation with 1,801 years, which we split into 86-year-long members to form a similar ensemble (our conclusions are unchanged if we instead concatenate each of the multi-member ensembles into a single long member). To ensure that the anomaly reference periods are handled equally, we

remove the mean in each ensemble member from 1961 to 1990 (for the PI Control "ensemble", we remove the mean of the $41^{st}$ to $70^{th}$ values in each member). To ensure that all datasets reflect only the internal component of climate variability, we remove the LENS historical ensemble mean time series from the LENS historical ensemble (to form LENS HI) and from the two reconstruction ensembles. The LENS historical ensemble mean time series represents the variability resulting from historical external forcing (Kay et al., 2015). All datasets used for processing and the

calculation are shown in Figure A4.

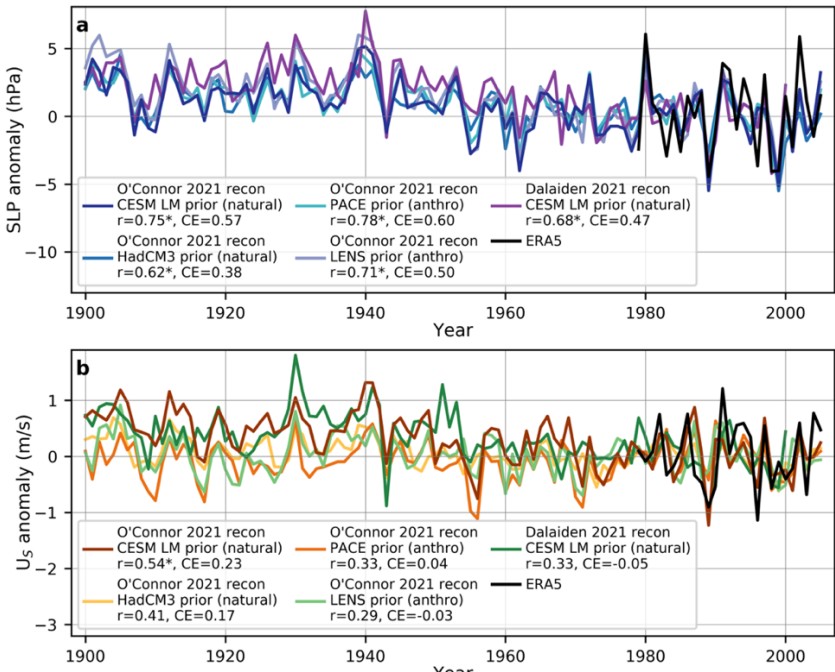

**Figure A1.** SLP and U$_S$ anomalies in the four reconstructions from O'Connor et al. (2021a), the reconstruction from Dalaiden et al. (2021), and ERA5, all averaged over the ASE continental shelf break box shown in Figure 1. For each reconstruction, the climate model simulation used as the data assimilation prior (and whether it include natural or anthropogenic forcing) is shown in the legend. The two reconstructions used in this study are the O'Connor et al. (2021a) CESM LM prior reconstruction (referred to as the natural-prior reconstruction) and the tropical Pacific pacemaker ('PACE') prior reconstruction (referred to as the anthro-prior reconstruction). Correlation and CE values compared to ERA5 for the period 1979-2000 are also shown in the legend. The anomaly reference period is 1979-2000 (the period of overlap).

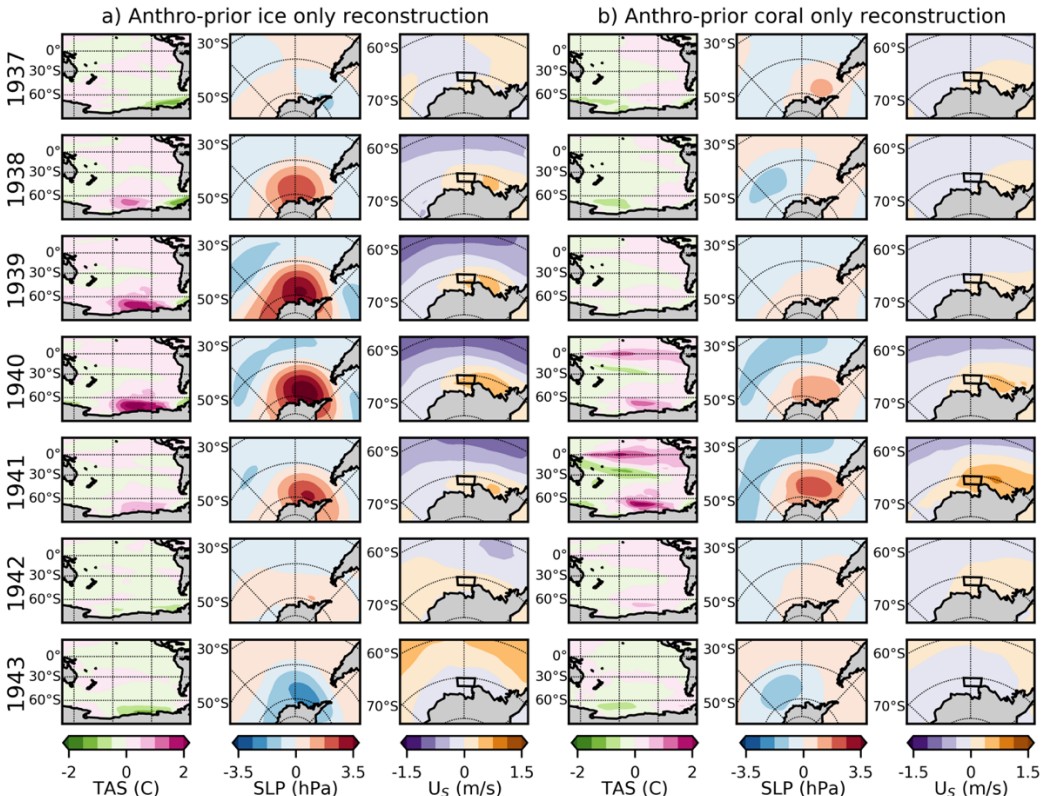

**Figure A2.** Same as in Figure 4 but using the tropical Pacific pacemaker ensemble as the prior (the anthro-prior).


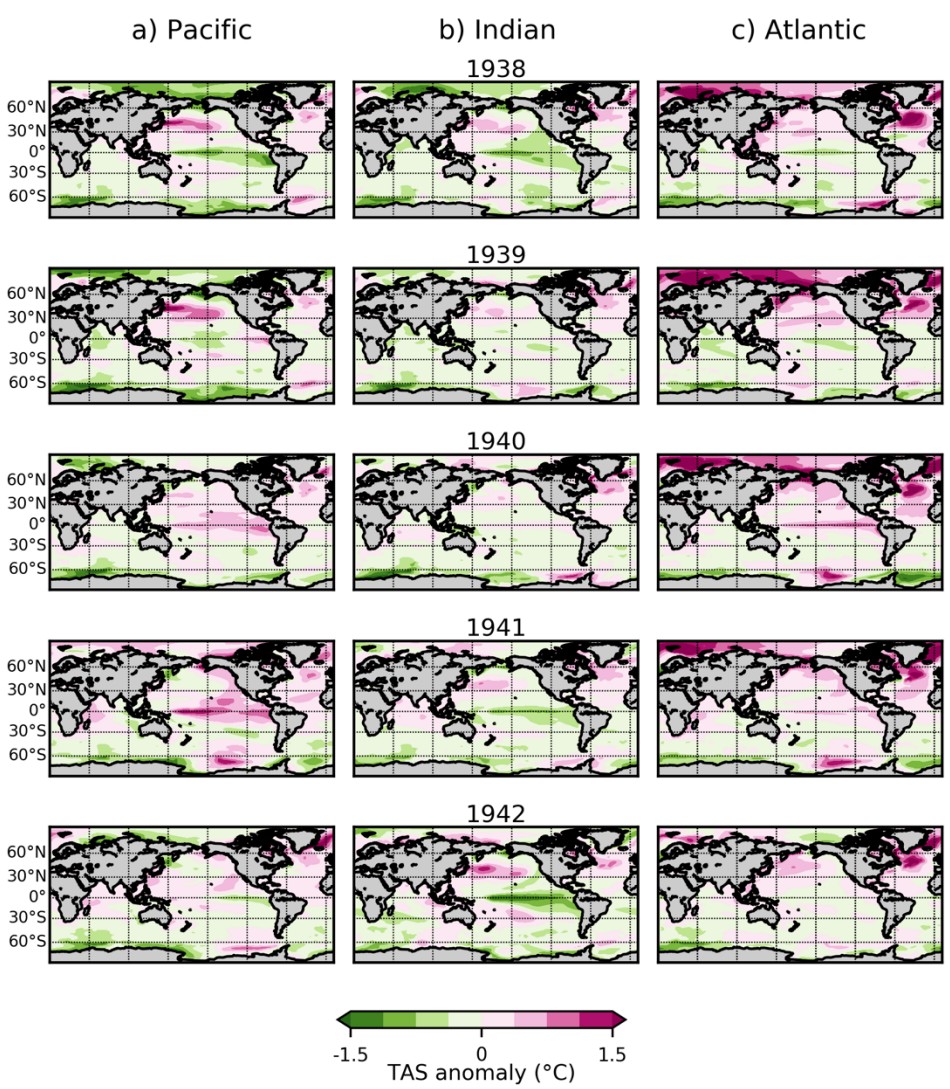


**Figure A3.** TAS anomalies in the ensemble mean of the (a) Pacific, (b) Indian, and (c) Atlantic Ocean pacemaker simulations from 1938 to 1942. Anomaly reference period is 1961-1990.


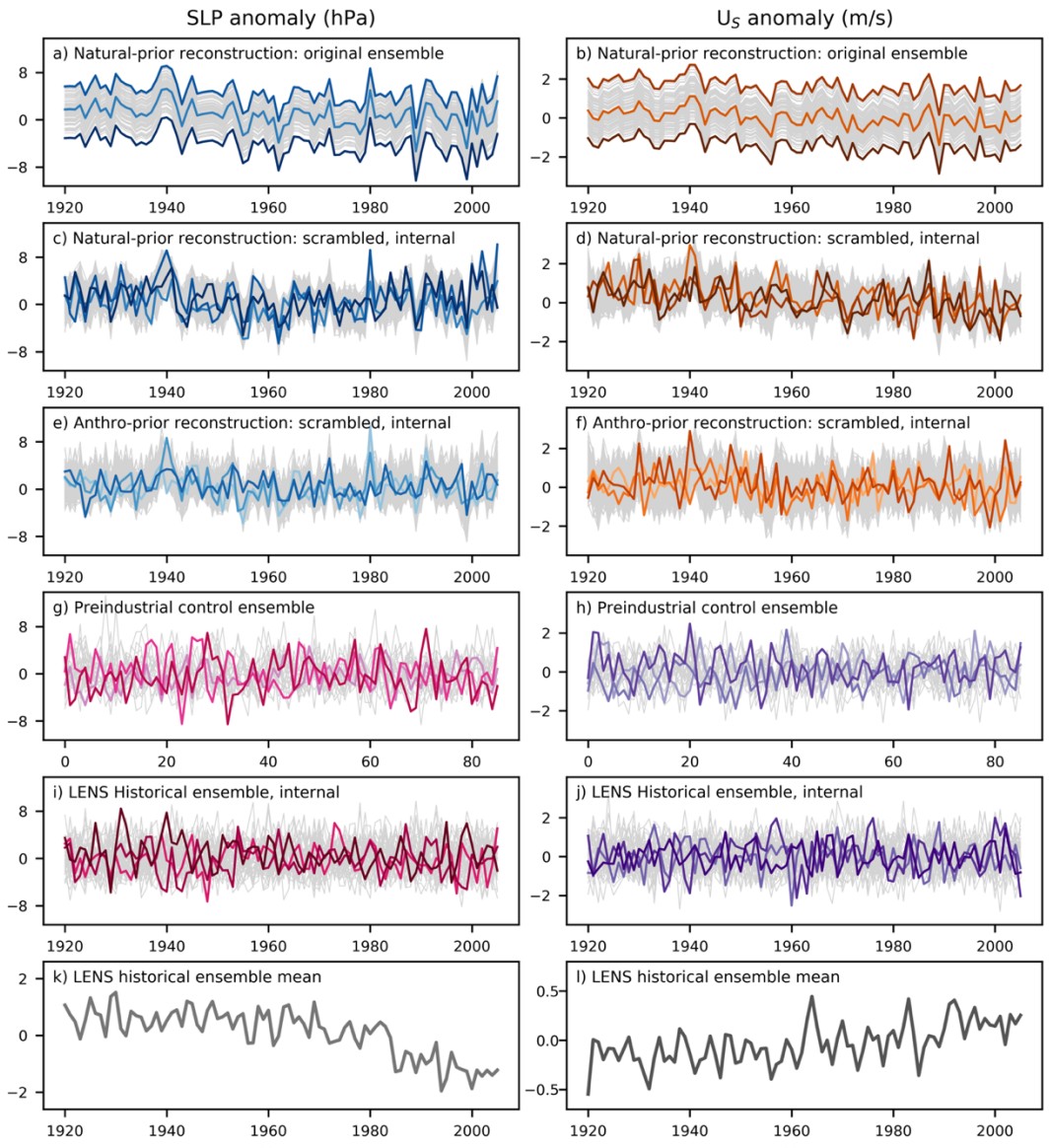

**Figure A4.** All timeseries used in the calculation of the number of occurrences of the 1940s event for SLP (left column) and $U_S$ (right column): (a, b) the original 100 ensemble members of the natural-prior reconstruction; (c, d) the internal component of the natural-prior reconstruction ensemble, after scrambling the members; (e, f) the internal component of the anthro-prior reconstruction ensemble, after scrambling the members; (g, h) the Preindustrial Control climate model simulation as an ensemble of 86-year-long chunks; (i, j) the internal component of the LENS historical simulation; (k, l) the LENS historical model ensemble mean. All timeseries are in anomalies relative to 1961 to 1990 (or the 41st to 70th values in the Preindustrial Control ensemble). Only three members in each ensemble of data are plotted in color to highlight the variability of each member (the ensemble members containing the minimum, maximum, and median value during 1940 -- or the 20th values in the Preindustrial Control -- are selected); all other members are plotted in gray.

**Acknowledgements**

G.K.O. was supported by the National Science Foundation Graduate Research Fellowship Program and the University
        of Washington Peter Misch Fellowship. This study was also partly supported by NSF grants 1602435, 1841844, and
        2045075 to E.J.S.

**Author Contributions**

G.K.O, P.R.H, and E.J.S. conceived the study. G.K.O. led the formal analysis and investigation with supervision from
        P.R.H and E.J.S. All authors contributed to the methodology, interpretation of results, and writing of the manuscript.

**Code and Data Availability**

        All code used to generate the analysis and figures in this study will be publicly available at
https://github.com/goconnor6 prior to publication. The code used to generate the single proxy reconstructions
        presented here is available at https://github.com/modons/LMR with documentation at
        https://atmos.washington.edu/~hakim/lmr/docs. The single proxy reconstructions are archived at Zenodo at
        https://zenodo.org/record/8007655. The other reconstruction data are available at O'Connor et al. (2021a) and
        Dalaiden et al. (2021). The simulations from CESM1 are available on the NCAR Climate Data Gateway. The ERA5
datasets are available at https://www.ecmwf.int/en/forecasts/datasets.

**Competing Interests**

The authors declare that they have no conflict of interest.

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
