# Peer review of "Characteristics and rarity of the strong 1940s westerly wind event over the Amundsen Sea, West Antarctica"

_The Cryosphere, 2023_

## Author Comment (AC1)

Reviewer Comment 1

Review comments for "Drivers and rarity of the strong 1940s westerly wind event over the Amundsen Sea, West Antarctica" by O'Connor et al. (tc-2023-16).

This study uses paleoclimate reconstructions to investigate atmospheric events in 1940s, which is supposed to be a trigger for the West Antarctic ice sheet retreat through wind-driven oceanic ice-shelf melting. The results demonstrate that the 1938-1942 anticyclonic anomalies and the associated westerly wind anomalies in the West Antarctic region can not be explained only by the El Niño event in 1940-1942, and thus other factors contributed to the atmospheric event. Furthermore, this study quantifies how rare this phenomenon is by comparing the characteristics (magnitude and persistent length) to results from several climate model simulations. In atmospheric reanalysis datasets that are widely available today (e.g., ERA5), atmospheric fields prior to 1979 are not strongly constrained by the data, and therefore, there are large uncertainties. However, it is important to know the long-term atmospheric variability for understanding climate components such as the Antarctic ice sheet and ocean, which have a long memory. Therefore, it is vital to understand the atmospheric conditions in the first half of the 20th century, and the atmospheric reconstructions by paleoclimate data in this study are very useful for this purpose. Thus, I believe that the purpose of this study is to the scope of The Cryosphere. However, I have some suggestions about the presentation and conclusions (see Major comments). I hope my comments will be helpful for the authors to improve the manuscript.

We thank the reviewer for their positive comments and constructive feedback.

Major comments

1. Motivation and Conclusion.

The manuscript starts with a motivation of understanding the retreat of the West Antarctic Ice Sheet in the 1940s due to the increased heat transport of Circumpolar Deep Water onto the continental shelf regions driven by the westerly wind anomalies in the 1940s (e.g. L10-14, and L26-43). However, the conclusion ends with the following sentences "Our results suggest that the 1940s event was probably not unprecedented in the Holocene. However, if the event were superimposed on favorable oceanic or glaciological conditions, or followed by anthropogenically forced trends, the event may have played a role in initiating ice loss. Ocean simulations forced by realistic climate histories, and continued direct observations in the field, are needed to better constrain the mechanisms responsible for glacier retreat in West Antarctica.", meaning that the westerly wind anomaly associated with the anticyclonic pressure anomalies is not a main driving force for the enhanced ice-shelf basal melting and the subsequent ice-sheet retreat. It was very confusing for me. Of course, there is no problem in describing the relations of the atmospheric change with ice sheets and oceans in the introduction and discussion, but motivating with understanding ice-sheet retreat and the causal ocean change does not seem to be suitable for this study. In fact, there is little analysis or discussion of ice sheets and oceans. Perhaps it would be better to structure the manuscript to focus more on items of atmospheric science.

We agree that this study about the 1940s winds does not directly address ice sheet and ocean changes near West Antarctica, since our focus is on the atmospheric components. However, the 1940s winds in this region are of interest to readers of The Cryosphere primarily because of their potential role in initiating the rapid ongoing glacier retreat in West Antarctica; our study therefore must make that motivation clear in the introduction and conclusion. How the atmospheric perturbation can explain

glacier retreat in West Antarctica is a key outstanding question in cryospheric science, and this study investigates a leading hypothesis on a possible atmospheric perturbation. We would like to clarify that we conclude that the 1940s event westerly event could indeed be the trigger of glacier retreat in the ASE (L495), but that additional factors are a necessary part of the narrative (suggested in L499-506), based on our finding that the event may have happened hundreds of times in 10kyr of internal climate variability. Our findings add novel constraints to the 1940s narrative and can be used to inform future ocean modelling and ice sheet modelling studies that can bridge the remaining gaps (L528). We will modify the introduction and discussion in the revised manuscript to clarify these points.

2. The expression about Holocene.

This study uses ensemble results from pre-industrial and 20th-century experiments to assess the rarity of the target atmospheric variability in the 1940s. In the manuscript, the authors discuss how many times such the atmospheric event occurs in 10,000 years, a comparable length to the Holocene. I think it's very misleading to refer to them as the Holocene probability. The Holocene has a similar land-ocean distribution to the present, but the other forcing of solar radiation and the freshwater cycle through ice sheets was very different from the present (e.g, "Holocene climate variability" by Mayewski et al. 2004). Throughout the manuscript, the phrases of the Holocene should be rephrased/removed if the characteristics of the Holocene are not taken into account in the climate simulations.

We thank the reviewer for raising this valid point. We will remove the term "Holocene" from the results and clarify that the frequency calculation is an estimate of occurrences per 10kyr. We will leave the implications of this frequency calculation for Holocene climate variability in the discussion, but we will add statements regarding the caveats associated with a comparison to the Holocene, such as those suggested by the reviewer.

3. Sentences about the rarity (Section 4)

What is the threshold between common and uncommon. Although the probability of occurrence is examined in Section 4, it seems strange that the results ("common" and "uncommon") can be changed simply by changing the length of the period.

The main purpose of our rarity analysis is to quantify the frequency of the event as a return period, which can be reported on different timescales simply to make the results more digestible. For example, 200 occurrences per 10kyr is more easily digested as 2 per century. We have not explicitly defined a threshold for "common" and "uncommon", as it would be arbitrary.  "Common" and "uncommon" are used in the text for qualitatively interpreting the frequency calculation in the context of the narrative that the ice was relatively stable for the last ~10kyr—not for making conclusions about statistical significance. We use sigma levels to determine a statistical threshold of the significance of the event (if the event magnitude exceeds 2 sigma, it is a statistically significant event in the context of 10kyr of internal climate variability). A statistically significant event does not necessarily mean the event was exceptional enough to explain the ocean changes needed to trigger glacier retreat in West Antarctica. Therefore, we present the results of our analyses (1) as a return period to be interpreted qualitatively in the narrative of glacier retreat, and (2) as an event that is either statistically significant or not, in the context of internal climate variability. We will revise section 4 to ensure that this is clearly stated.

4. The other variability (local variability or response to tropical forcing outside the Pacific, L296-297).

After reading the manuscript, I ended up not knowing what "the other variability" was. It seems essential to find out "the other variability" in the paleoclimate reconstructions (local response or Tropical Atlantic wave train in Li et al. 2021?). It seems to me that showing the extent of influence of the ice core data in the paleoclimate reconstruction would be helpful in understanding the variability and the pattern.

The ice core-only reconstructions presented in the manuscript is one approach at investigating the influence of the ice core data in the reconstruction. This is because the variability in the reconstruction is derived from the proxy data (L141): at each grid point, the climate model prior time series is simply a flat line which is updated as each proxy record is assimilated, resulting in the final reconstructed time series shown in this study. Thus, the variability shown in the ice core-only reconstruction but not in the coral-only reconstruction (Figure 4) shows the influence of each proxy type in the all-proxy reconstruction.

Because the other reviewers also raise the question of what the other sources of variability may be, we will revise the text to include results from two additional ensembles of pacemaker simulations. These are simulations from CESM1 that are constrained to follow observed SSTs from the North Atlantic Ocean or the Indian Ocean (Yang et al., 2020; using the same approach as the tropical Pacific pacemaker simulation already used in our study). The ensemble mean of the simulations reflects the mean response to observed variability in the respective restoring SST ocean basin, so the results from these simulations provide an avenue for investigating alternative sources of variability in the ASE during the 1940s.

The SLP anomalies in the ensemble means of these simulations suggest that the high-pressure anomalies in the ASE are associated with Indian Ocean variability in 1938 and 1939, by North Atlantic variability in 1939, and by Pacific variability in 1940 and 1941 (see Figure 1 below). This suggests that the overall 1940s anomaly was the result of a confluence of different climate modes operating in succession over the Amundsen Sea. The results for zonal winds are less conclusive but suggest that the westerly anomalies in 1939 may be related to a North Atlantic warm anomaly. North Atlantic warming may also contribute to the anomalies in 1940 and 1941, but it is difficult to isolate the variability between the Pacific and the Atlantic (i.e., the North Atlantic warming that occurred in 1940/41 may be a response to the El Niño). Although these simulations are associated with large uncertainties, they provide additional insight into the potential sources of other variability. We note that the results from the Indian and Atlantic pacemaker simulations may be considered reliable only if they are broadly consistent with tropical Pacific variability, which is indeed the case for the Indian pacemaker simulation in 1938 and 1939, and for the Atlantic pacemaker from 1939 to 1941 (see Figure 2 below). We will therefore incorporate these results as figures in the appendix of our revised manuscript, along with new text to section 3.2 describing the above inferences.

[Figure]

Figure 1. SLP and Us anomalies in annually resolved pacemaker simulations (using the ensemble mean) with restoring sea surface temperatures from different basins (Pacific, Indian, and Atlantic Oceans) from 1938 to 1941. Anomaly reference period is 1961 to 1990.

[Figure]

Figure 2. TAS anomalies from 1938 to 1942 in the CESM1 pacemaker ensemble means. Anomaly reference period is 1961 to 1990.

5. Readability (assumed readers)

While this manuscript may be understandable to researchers focusing on West Antarctica, I found it difficult to read for a broad general audience in The Cryosphere (including me). It would be easier to

read if there were two large panels (as Fig 1) showing the mean atmospheric fields and the anomaly fields in 1940s. I hope that the locations of ice core and coral records are also plotted.

Only the anomalies in the 1940s have been reconstructed. We will add maps of mean SLP and $U_S$ from ERA5 to Figure 1 to provide an additional point of reference for readers unfamiliar with this region. We will add locations of the proxy data to this plot.

Specific comments

6. Appendices

Since the appendices have only one paragraph, I suggest the author to include them in the main text.

We found that the text in the appendices greatly disrupted the flow of section 4, which already contains many details on the frequency calculation. We think these additional details are only of interest to a small group of readers interested in the precise details of the calculation, and so do not warrant their own sub-sections in the main text. Therefore, we have decided to keep this text in the appendices.

7. Fig. 2

All the ensembles hardly cross each other and remain parallel. What determines this variance? I think the scrambled reconstruction has the same variance. Is it correct?

The original ensemble members indeed have the same interannual variability and differ only by their means, which are based on a random draw from the climate model prior. The variance in each member is derived from the proxy data using a Bayesian approach in which the proxy observations are used to update the initial estimate from the prior. Because each original ensemble member is identical except for the mean, we scramble the ensemble members to generate unique realizations of the ensemble that keeps the same variance as the original ensemble. This is explained at a high level in the methods (L146-162) and in more detail in Appendix A. More details on the data assimilation method can be found in Hakim et al., 2016. We will revise the text to clarify these points.

8. Fig. 3

Panels in different columns use different projections and spatial domains. Could you please use the same spatial domain at least for SLP and Us? Furthermore, showing bathymetric features (e.g., 1000- and 3000-m depth contour as the representative of shelf break position) is helpful.

We will revise the maps to have more consistent spatial domains and add a contour of the shelf break position.

New References

Yang D., Arblaster, J.M., Meehl, G.A., England, M.H., Lim, E.-P., Bates, S., Rosenbloom, N.: Role of tropical variability in driving decadal shifts in the Southern Hemisphere summertime eddy-driven jet, Journal of Climate, 10.1175/JCLI-D-19-0604.1, 2020.

---

## Author Comment (AC2)

Reviewer Comment 2

O'Connor et al use reconstructed winds and sea level pressure over the Amundsen Sea to quantify the rarity of westerly wind anomalies that occurred between the late 1930s and the early 1940s. These winds anomalies have been associated with anomalous ocean heat transport toward Amundsen Sea glaciers, possibly favouring the glacial retreat in the 1940s inferred from sediments. The authors use proxy-based reconstructions that have been shown to well reproduce present atmospheric variability. I found the analysis accurate and the results very interesting as they provide 1) new information on the mechanisms driving atmospheric variability in the Amundsen Sea and 2) useful insight on the role of atmospheric variability in changes of outlet glaciers in the Amundsen Sea. In particular, the authors find that this event was rare on centennial time scale, but common on millennial time scale. This suggests that multiple drivers beyond anomalous local winds in the 1940s acted to initiate the retreat of glaciers in the Amundsen Sea. I have a couple of major comments below and few minor suggestions.

We thank the reviewer for their time and for their positive and constructive comments.

**Major Comments**

- Line 185-200: you refer here to internal climate variability. But is this internal climate variability reflecting the "modern climate variability" or does it include evolving climate variability over several thousands of years? e.g. is ENSO variability the same over the past 10000 years?

- Section 4: Is the rarity of the event relative to the last 10000 years or to a "repetition" of the present state? I think this should be clarified because this might have implications on connections with the stability of the WAIS over the past 10K years and the initiation of the retreat in the last century.

Our frequency calculation per 10kyr is based on the CESM1 LENS preindustrial and historical (internal component only) simulations, so the calculation reflects repeated modern climate variability. We recognize that this is an imperfect analogy to Holocene variability. Building on our response to reviewer #1, we will revise the text to ensure that our results are restricted to the terminology "per 10kyr of internal climate variability" and only include the analogy to the Holocene in the discussion, with some added text about the caveats of this analogy. These caveats will include the possibility that ENSO variability throughout the Holocene is different than modern climate, and that forcings from solar radiation and the freshwater cycle are different, as suggested by reviewer #1. There is paleo-evidence that ENSO was highly variable throughout the Holocene, including variability resembling the 20[th] century (Cobb et al., 2013). There is also paleo-evidence that ENSO variability was damped during parts of the Holocene (e.g., Conroy et al., 2008), which would suggest that our calculations represent a conservative estimate of the rarity of the 1940s event. We will note these caveats in our revised manuscript.

**Minor comments**

- Line 30. I like this introductory paragraph and find useful to put things into context. However, I feel the wording "There is evidence that these glaciers have been relatively stable for the last ~10,000 years (Larter et al., 2014), which implies that a change in ocean circulation, and a corresponding increase in heat delivery to the ice shelves, must have occurred to initiate the current stage of retreat" to be a bit too strong here. Could a slow change in ocean forcing and/or surface mass balance be part of the story?

We will modify this statement by changing it to: "some change in ocean circulation… likely occurred". We would also like to clarify three distinct components of the narrative of glacier retreat in the ASE: (1) the historical trigger of the retreat, (2) the underlying cause of the ongoing retreat, and (3) the ongoing retreat itself. The focus of this paper is on the trigger of retreat. The reviewer's suggestion about a change in ocean forcing or surface mass balance relates to the underlying causes of ongoing retreat, which could include slow changes in surface mass balance, slow changes in ocean forcing (Hillenbrand et al., 2017), or anthropogenic forcing. The ongoing retreat itself is governed by ice/ocean feedbacks (e.g., Joughin et al., 2014; Holland et al., 2023) of a greater scale than the changes from surface mass balance (Shepherd et al., 2002). We will clarify these points in our revised manuscript.

- Are winds in the Amundsen Sea important only for ice shelf basal melting? What about carbon uptake, ecosystems, sea ice etc.? how a rare wind event would impact other components of the system? Might be worth adding a few lines, given that this manuscript does not focus on ice shelves.

  Yes, winds in this region are important for additional reasons including carbon uptake, primary productivity, and sea ice extent. We will add a statement regarding these other motivators to learn about the 1940s wind event.

- There are also other mechanisms (beyond shelf break winds) that have been proposed to potentially explain changes in ice shelf basal melting in the Amundsen Sea, including surface buoyancy forcing and remote forcing (Ross Gyre and melting of glaciers in the Western Peninsula). I would suggest to either mention that other mechanisms have been proposed and therefore this new study can provide new information on potential processes, or simplify and shorten the text highlighting that rare winds events can affect heat delivery to ice shelves and potentially their stability. Given the strong motivation and evidence based on sediments and ocean dynamical studies, the first option might be more appropriate.

  Remote drivers may play a key role in driving CDW transport in the ASE, such as suggested by Nakayama et al., 2018. We refer to this in the discussion in L487, noting that it is a much less developed area of research but requires further investigation. We will add a statement including alternative mechanisms related to increased freshwater flux such as the buoyancy mechanism (e.g., Webber et al., 2019) and melt originating in the Antarctic Peninsula (Flexas et al., 2022), which could have played a role in explaining why the ice failed to recover after the initial perturbation in the 1940s.

- Line 130: "Magnitude" of what? Please specify.

  We will specify in L130 that we are referring to the magnitude of the zonal wind anomalies.

- Line 215: "SAT" for surface air temperature?

  "SAT" and "TAS" are both commonly used for 2m air temperature, so we will keep it as "TAS".

- Line 465. I would provide a very short summary of the key results at the beginning of the Discussion.

We will add a short summary of the major findings of this study to the discussion section.

- Line 500-505. The discussion here is very important as it highlights that many mechanisms are at play. As suggested before, some of these processes should be mentioned in the Introduction. Something that could be also discussed a bit more is the role of the IPO in all this. A recent study by Vance et al (https://doi.org/10.1038/s43247-022-00359-z ) suggests IPO anomalies in the 20th century. Would this be important for the 1938-1942 event? Or for trend/interdecadal variability over the reconstructed period?

  Holland et al. (2022) investigate the role of the IPO on atmospheric variability and century-scale trends in the ASE. The IPO is a dominant influence on interdecadal timescales, but for interannual timescales which are more relevant to the 1940s wind event, ENSO is the dominant driver and at least partly explains the anticyclonic anomalies shown in the reconstructions. We will note this in our revised manuscript.

- Discussion and conclusion. Similar to the introduction, would an event like this affect other components of the system?

  It is possible that this large westerly event could have affected other components of the system, such as sea ice and therefore affecting biological processes and carbon uptake. We will add a brief statement on this to the discussion section.

New References

Cobb, K. M., Westphal., N., Sayani, H. R., Watson, J. T., Di Lorenzo, E., Cheng, H., Edwards, R. L., Charles, C. D.: Highly variable El Niño-Southern Oscillation throughout the Holocene, Science, 339, 67-70, 10.1126/science.1228246, 2013.

Conroy, J. L., Overpeck, J. T., Cole, J. E., Shanahan, T. M., Steinits-Kannan, M.: Holocene changes in eastern tropical Pacific climate inferred from a Galapagos lake sediment record, Quat. Sci. Rev., 27 (11-12), 1166-1180, 10.1016/j.quascirev.2008.02.015, 2008.

Flexas, M. M., Thompson, A. F., Schodlok, M. P., Zhang, H., and Speer, K.: Antarctic Peninsula warming triggers enhanced basal melt rates throughout West Antarctica, Science Advances, 8 (31), eabj9134, 10.1126/sciadv.abj9134, 2022.

Joughin, I., Smith, B. E., Medley, B.: Marine Ice Sheet collapse potentially under way for Thwaites Glacier Basin, West Antarctica, Science, 344, 735-738, 10.1126/science.1249055, 2014.

Holland, P. R., Bevan, S. L., Luckman, A. J.: Strong ocean melting feedback during the recent retreat of Thwaites Glacier, Geophys. Res. Lett., 50(8), e2023GL103088, 10.1029/2023GL103088, 2023.

Shepherd, A., Wingham, D., Mansley, J.A.: Inland thinning of the Amundsen Sea sector, West Antarctica. Geophys. Res. Lett., 29(10), 1364, 10.1029/2001GL014183, 2002.

Webber, B. G. M., Heywood, K. J., Stevens, D. P., Assmann, K. M.: The impact of overturning and horizontal circulation in Pine Island Trough on ice shelf melt in the eastern Amundsen Sea, Am. Met. Soc., 49, 63-83, 10.1175/JPO-D-17-0213.1, 2019.

---

## Author Comment (AC3)

Reviewer Comment 3

This study uses paleoclimate reconstructions to assess the drivers and rarity of the 1940s events that led to large ice shelf melting and glacier retreat in the ASE. These events were rare due to their large magnitude and duration. Local forcings in combination with a major ENSO led to the 1940s events, which are rare on centennial timescales but uncommon on millennial timescales. The paper is written in an elegant way, with clear messages and figures. The subject is of interest to glaciologists and oceanographers studying West Antarctica, and I recommend publication in The Cryosphere. Below are some comments and suggestions that might help to improve the manuscript.

We thank the reviewer for their positive review and helpful comments.

Major:

1) The authors say they evaluated the rarity of the event in the Holocene, but their simulation goes back to ~2000 yr (lines 192-194). Although this is part of the Holocene period, it does not cover the full Holocene or the Holocene conditions. I need a better explanation of why they claim the analysis covers the Holocene. Or, the authors could change Holocene to a simple "10kyr period"?

Following our response to the other reviews, we will revise the text to note the uncertainty associated with repeating the LENS simulations as an analogy for the Holocene.

2) Section 3.2: When I read "Drivers" (also in the title), I thought the authors would track down the origin of the rare 1940s event. But I have the feeling that they don't. Instead, their assumption is that other events in combination with ENSO could trigger and amplify the 1940s event. My questions remained: How this event was generated? What (local conditions) triggered this event? Saying that the 1940s event was due to local drivers and not totally ENSO-related does not properly address the "Drivers" of the 1940s event, in my opinion. I'd like to see more analysis on this to properly address what led to the 1940s event. I think this is a key question that could benefit this paper to be a greater contribution to the glaciological/ocean/atmospheric community. If the authors decided to not track down the causes of the 1940s event, which I understand can be a lot of work, I'd recommend avoiding using "Drivers". However, I think understanding the local conditions could be a great addition to the paper.

Following reviewer #1's similar comment, we will add some new analyses, figures, and text to investigate the potential additional drivers of the 1940s event. This will include analyses from the Atlantic and Indian ocean pacemaker simulations. Please refer to our explanation of this analysis in our response to reviewer #1.

We note that this does not address the drivers of local variability in the Amundsen Sea, which have been investigated in previous studies (e.g., Raphael et al., 2016; Goyal et al., 2021). We will remove the use of the word "Drivers" in our manuscript title and revise our use of the word "drivers" throughout the text.

Minor:

- Line 294: Figure 5 instead of Figure 4?

Yes, that is a typo. We will change the reference to Figure 5 in the revised manuscript. We thank the reviewer for catching this.

- Line 348: Figure 6e?

Yes, that is also a typo. We will change the text in the revised manuscript.

- Why the authors use 10kyr most of the time, but sometimes 10ka?

We will change all cases of "10ka" to "10kyr".

- In the authors' opinion, what is most relevant for the ice shelf melting that occurred in the ASE: duration or magnitude of the 1940s event?

We suggest that it's the combination of the two (i.e., the time-integral of the anomaly) that caused a significant perturbation to the ice shelves. Modern observations and the reconstructions show that similar magnitude events have occurred several times before in only the last several decades, and similarly persistent events have occurred several times before—but never one that is both large in magnitude and duration. However, ocean simulations that investigate the sensitivity of CDW transport and ice shelf melt to wind events with different characteristics are needed to answer this.

New References

Goyal, R., Jucker, M., Gupta, A. S., England, M. H.: Generation of the Amundsen Sea Low by Antarctic orography, Geophys. Res. Lett., 48, e2020GL091487, 10.1029/2020GL091487, 2021.

Raphael., M. R., Marshall. G. J., Turner, J., Fogt, R. L., Schneider, D., Dixon, D. A., Hosking, J. S., Jones, J. M., Hobbs, W. R.: The Amundsen Sea Low: variability, change, and impact on Antarctic climate, Am. Met. Soc., 97(1), 111-121, 10.1175/BAMS-D-14-00018.1, 2016.

---

## Author Response (AR1)

**Reviewer Comment 1**

Review comments for "Drivers and rarity of the strong 1940s westerly wind event over the Amundsen Sea, West Antarctica" by O'Connor et al. (tc-2023-16).

This study uses paleoclimate reconstructions to investigate atmospheric events in 1940s, which is supposed to be a trigger for the West Antarctic ice sheet retreat through wind-driven oceanic ice-shelf melting. The results demonstrate that the 1938-1942 anticyclonic anomalies and the associated westerly wind anomalies in the West Antarctic region can not be explained only by the El Niño event in 1940-1942, and thus other factors contributed to the atmospheric event. Furthermore, this study quantifies how rare this phenomenon is by comparing the characteristics (magnitude and persistent length) to results from several climate model simulations. In atmospheric reanalysis datasets that are widely available today (e.g., ERA5), atmospheric fields prior to 1979 are not strongly constrained by the data, and therefore, there are large uncertainties. However, it is important to know the long-term atmospheric variability for understanding climate components such as the Antarctic ice sheet and ocean, which have a long memory. Therefore, it is vital to understand the atmospheric conditions in the first half of the 20th century, and the atmospheric reconstructions by paleoclimate data in this study are very useful for this purpose. Thus, I believe that the purpose of this study is to the scope of The Cryosphere. However, I have some suggestions about the presentation and conclusions (see Major comments). I hope my comments will be helpful for the authors to improve the manuscript.

We thank the reviewer for their positive comments and constructive feedback which have improved our study.

Major comments

1. Motivation and Conclusion.

The manuscript starts with a motivation of understanding the retreat of the West Antarctic Ice Sheet in the 1940s due to the increased heat transport of Circumpolar Deep Water onto the continental shelf regions driven by the westerly wind anomalies in the 1940s (e.g. L10-14, and L26-43). However, the conclusion ends with the following sentences "Our results suggest that the 1940s event was probably not unprecedented in the Holocene. However, if the event were superimposed on favorable oceanic or glaciological conditions, or followed by anthropogenically forced trends, the event may have played a role in initiating ice loss. Ocean simulations forced by realistic climate histories, and continued direct observations in the field, are needed to better constrain the mechanisms responsible for glacier retreat in West Antarctica.", meaning that the westerly wind anomaly associated with the anticyclonic pressure anomalies is not a main driving force for the enhanced ice-shelf basal melting and the subsequent ice-sheet retreat. It was very confusing for me. Of course, there is no problem in describing the relations of the atmospheric change with ice sheets and oceans in the introduction and discussion, but motivating with understanding ice-sheet retreat and the causal ocean change does not seem to be suitable for this study. In fact, there is little analysis or discussion of ice sheets and oceans. Perhaps it would be better to structure the manuscript to focus more on items of atmospheric science.

Ice sheet and ocean changes near West Antarctica depend strongly on atmospheric forcing, so the 1940s winds in this region are of interest to readers of The Cryosphere, primarily because of their potential role in initiating the rapid ongoing glacier retreat in West Antarctica. Our study therefore must make that motivation clear in the introduction and conclusion. How the atmospheric perturbation can explain

glacier retreat in West Antarctica is a key outstanding question in cryospheric science, and this study investigates a leading hypothesis on a possible atmospheric perturbation. We state this in L95 of the tracked changes manuscript:

> "we use annually resolved proxy reconstructions and climate model simulations of surface pressure and winds to further investigate the hypothesis that a large atmospheric event in the ASE around 1940 forced the ocean-induced changes responsible for triggering glacier retreat. Specifically, we investigate the significance of the atmospheric component of the hypothesis."

We would like to clarify that we conclude that the 1940s westerly event could indeed be the immediate trigger of glacier retreat in the ASE, but that additional factors are a necessary part of the narrative, based on our finding that the event may have happened hundreds of times in 10kyr of internal climate variability. Our findings add novel constraints to the 1940s narrative and reveal new uncertainties associated with the narrative and require further investigation in future ocean modelling and ice sheet modelling studies. We revised the conclusions to clarify these points (L907 of the tracked changes manuscript):

> "Our rarity estimates show that the 1940s pressure and zonal wind anomalies are likely to occur tens to hundreds of times in 10kyr or internal climate variability, suggesting that the event is likely not unprecedented in the Holocene, and it may not be particularly exceptional. Our results reveal new uncertainties associated with the narrative that the westerly event may have triggered West Antarctic glacier retreat. The 1940s event may have been the initial atmospheric perturbation, however given the estimated likelihood of this type of event, other factors are a necessary component to the narrative. It is unlikely that the 1940s atmospheric perturbation alone can explain the current stage of glacier retreat in the ASE. We suggest that if the event were superimposed on favorable oceanic or glaciological conditions, or followed by anthropogenically forced trends, the event may have played a role in initiating ice loss. Ocean simulations forced by realistic climate histories, and continued direct observations in the field, are needed to better constrain the mechanisms responsible for glacier retreat in West Antarctica."

2. The expression about Holocene.

This study uses ensemble results from pre-industrial and 20th-century experiments to assess the rarity of the target atmospheric variability in the 1940s. In the manuscript, the authors discuss how many times such the atmospheric event occurs in 10,000 years, a comparable length to the Holocene. I think it's very misleading to refer to them as the Holocene probability. The Holocene has a similar land-ocean distribution to the present, but the other forcing of solar radiation and the freshwater cycle through ice sheets was very different from the present (e.g, "Holocene climate variability" by Mayewski et al. 2004). Throughout the manuscript, the phrases of the Holocene should be rephrased/removed if the characteristics of the Holocene are not taken into account in the climate simulations.

We thank the reviewer for raising this valid point. We have removed the term "Holocene" from the results and clarify that the frequency calculation is an estimate of occurrences per 10kyr. We note the implications of using this calculation for an analogy to Holocene climate variability in the methods (L249 of the tracked changes manuscript):

> "The simulations allow us to quantify the rarity of the 1940s event relative to 10kyr of internal pre-industrial climate variability, providing an estimate for the significance of the event relative to the Holocene. We note that these simulations are an imperfect analogy to Holocene climate variability; we discuss the caveats associated with this analogy in the discussion section."

And in the discussion (L851):

> "We note that the internal component of the LENS simulations are an imperfect analogy to the Holocene, which experienced differences in insolation, freshwater inputs, and potentially ENSO variability (e.g., Mayewski et al., 2004). However, the influence of these differences on climate variability near West Antarctica is highly uncertain; e.g., there is no clear consensus on whether Holocene ENSO variability was similar (e.g., Cobb et al., 2013) or damped (e.g., Conroy et al., 2008; Grothe et al., 2020) relative to modern climate. Thus, the results presented here are associated with uncertainties but provide an estimate of the significance of the 1940s event relative to the best available simulations."

3. Sentences about the rarity (Section 4)

What is the threshold between common and uncommon. Although the probability of occurrence is examined in Section 4, it seems strange that the results ("common" and "uncommon") can be changed simply by changing the length of the period.

To clarify our qualitative classification of common vs. uncommon, we have added three classifications based on different criteria: "unprecedented", "exceptional", and "relatively uncommon", based on the bellow criteria added to L451 of the tracked changes manuscript:

> "We use the sigma level to quantify the statistical significance of the event. Because a statistically significant event does not necessarily explain the start of glacier retreat, we use the frequency estimates to evaluate the narrative that the 1940s event may explain the start of glacier retreat in West Antarctica. If the 95% confidence interval of the frequency includes 1 per 10kyr, we fail to reject the hypothesis that the event is unprecedented. If the confidence interval includes <20 events per 10kyr , we classify the event as "exceptional". If the event fails to meet these criteria but is a 2-sigma event, we classify it as "relatively uncommon". We note that the number 20 for "exceptional" is relatively arbitrary; it is simply used to define a threshold for qualitative descriptions of the results, and our main conclusions are insensitive to this precise choice (i.e., the conclusions are unchanged if choose 5 or 50)."

We made additional revisions to the remaining text in section 4 to ensure that this classification is clearly followed and to improve the readability of the text. The changes can all be seen in the tracked changes manuscript.

4. The other variability (local variability or response to tropical forcing outside the Pacific, L296-297).

After reading the manuscript, I ended up not knowing what "the other variability" was. It seems essential to find out "the other variability" in the paleoclimate reconstructions (local response or Tropical Atlantic wave train in Li et al. 2021?). It seems to me that showing the extent of influence of the

ice core data in the paleoclimate reconstruction would be helpful in understanding the variability and the pattern.

The ice-core-only reconstructions presented in the manuscript is one approach to investigating the influence of the ice core data in the reconstruction. This is because the variability in the reconstruction is derived from the proxy data (L183): at each grid point, the climate model prior time series is zero, and updated as each proxy record is assimilated, resulting in the final reconstructed time series shown in this study. Thus, the variability shown in the ice-core-only reconstruction but not in the coral-only reconstruction (Figure 4) shows the influence of each proxy type in the all-proxy reconstruction.

Because the other reviewers also raise the question of what the other sources of variability may be, we have revised the text to include results from two additional ensemble pacemaker simulations in Figures 6 and A3. These are simulations from CESM1 that are constrained to follow observed SSTs from the North Atlantic Ocean or the Indian Ocean (Yang et al., 2020; using the same approach as the tropical Pacific pacemaker simulation already used in our study). The ensemble mean of the simulations reflects the mean response to observed variability in the respective restoring SST ocean basin, so the results from these simulations provide an avenue for investigating alternative sources of variability in the ASE during the 1940s. We added the following text to section 3.2, L358, to explain these additional results:

> "To investigate additional sources of variability associated with the event in the ASE, we next evaluate two additional CESM1 pacemaker simulations: one constrained to follow observed SSTs from the North Atlantic, and one constrained to follow observed SSTS from the Indian Ocean (Yang et al., 2020).  Both simulations include historical external forcing. Like the tropical Pacific pacemaker simulation, the ensemble means of these simulations reflect the mean response to observed variability in the respective restoring SST ocean basin and external forcing. The SLP and US anomalies in all three pacemaker simulations are shown in Fig. 6. In 1939, high pressure anomalies are found in the Indian and Atlantic simulations; in 1940, high pressures are shown in the Pacific and Atlantic simulations; and in 1941, high pressures are shown in the Pacific and Atlantic simulations. The results for US are more subtle: weakly westerly anomalies are shown in the Atlantic pacemaker simulation from 1939 to 1941. These results suggest that the large anticyclonic anomalies shown in the reconstructions may be a result of a confluence of different climate modes operating in succession.
>
> The results from the Atlantic and Indian pacemaker simulations may be considered reliable only if they are broadly consistent with tropical Pacific variability, which is indeed the case for the Indian pacemaker simulations in 1938 and 1939, and for the Atlantic pacemaker simulation from 1939 to 1941 (TAS anomalies in all three pacemaker simulations are shown in Fig. A3). However, we note that it is difficult to isolate the variability between the Pacific and other ocean basins (i.e., the observed North Atlantic warming may be associated with the El Niño). Thus, while the results from these simulations are associated with large uncertainties, they provide insight into the potential sources of other variability that may explain the 1940s event, consistent with previous work (e.g., Okumura et al., 2012; Li et al., 2021). Combining the results from the single-proxy experiments, the tropical Pacific pacemaker simulation, and these two additional pacemaker simulations, the evidence suggests that the 1940s anti-cyclonic anomalies over the Amundsen Sea are associated with a combination of factors resulting in a potentially rare "perfect storm"."

Given the uncertainty associated with the other sources of variability, we have also changed the title of the study from "Drivers and rarity" to "Characteristics and rarity…" to weaken the emphasis on the drivers of the event.

5. Readability (assumed readers)

While this manuscript may be understandable to researchers focusing on West Antarctica, I found it difficult to read for a broad general audience in The Cryosphere (including me). It would be easier to read if there were two large panels (as Fig 1) showing the mean atmospheric fields and the anomaly fields in 1940s. I hope that the locations of ice core and coral records are also plotted.

Only the anomalies in the 1940s have been reconstructed, so we added maps of mean SLP and $U_S$ from ERA5 to Figure 1 to provide an additional point of reference for readers unfamiliar with this region. We also added locations of the proxy data to this plot.

Specific comments

6. Appendices

Since the appendices have only one paragraph, I suggest the author to include them in the main text.

We found that the text in the appendices greatly disrupted the flow of section 4, which already contains many details on the frequency calculation. We think these additional details are only of interest to a small group of readers interested in the precise details of the calculation, and so do not warrant their own sub-sections in the main text. Therefore, we have decided to keep this text in the appendices.

7. Fig. 2

All the ensembles hardly cross each other and remain parallel. What determines this variance? I think the scrambled reconstruction has the same variance. Is it correct?

Yes. The original ensemble members indeed have the same interannual variability and differ only by their means, which are based on a random draw from the climate model prior. The variance in each member is derived from the proxy data using a Bayesian approach in which the proxy observations are used to update the initial estimate from the prior. Because each original ensemble member is identical except for the mean, we scramble the ensemble members to generate unique time-series realizations of the ensemble that keeps the same variance as the original ensemble. This is explained at a high level in the methods (L197) and in more detail in Appendix A. More details on the data assimilation method can be found in Hakim et al., 2016.

8. Fig. 3

Panels in different columns use different projections and spatial domains. Could you please use the same spatial domain at least for SLP and Us? Furthermore, showing bathymetric features (e.g., 1000- and 3000-m depth contour as the representative of shelf break position) is helpful.

We experimented with remaking the figures using consistent spatial domains, but we ultimately decided to keep the domains as is for clarity; given how many panels are in each figure, we believe that the smallest domain appropriate for each variable is best for clearly visualizing the anomalies. The TAS domain includes the largest region so that the tropical Pacific warming signatures can be seen. The SLP domain includes the South Pacific so that the entire high-pressure region can be seen. The $U_S$ domain includes the smallest region because the westerly anomalies are focused over a smaller region, which are hard to see using the larger TAS or SLP domains. We hope that the addition of the maps in Figure 1d and 1e shown in the South Pacific assist with orienting the reader to the region. The location of the shelf break box used in our study is consistent with the location used by previous studies, which have shown the location relative to nearby bathymetric features (e.g., Holland et al., 2019).

Reviewer Comment 2

O'Connor et al use reconstructed winds and sea level pressure over the Amundsen Sea to quantify the rarity of westerly wind anomalies that occurred between the late 1930s and the early 1940s. These winds anomalies have been associated with anomalous ocean heat transport toward Amundsen Sea glaciers, possibly favouring the glacial retreat in the 1940s inferred from sediments. The authors use proxy-based reconstructions that have been shown to well reproduce present atmospheric variability. I found the analysis accurate and the results very interesting as they provide 1) new information on the mechanisms driving atmospheric variability in the Amundsen Sea and 2) useful insight on the role of atmospheric variability in changes of outlet glaciers in the Amundsen Sea. In particular, the authors find that this event was rare on centennial time scale, but common on millennial time scale. This suggests that multiple drivers beyond anomalous local winds in the 1940s acted to initiate the retreat of glaciers in the Amundsen Sea. I have a couple of major comments below and few minor suggestions.

We thank the reviewer for their time and for their positive and constructive comments which have improved the manuscript.

**Major Comments**

- Line 185-200: you refer here to internal climate variability. But is this internal climate variability reflecting the "modern climate variability" or does it include evolving climate variability over several thousands of years? e.g. is ENSO variability the same over the past 10000 years?

- Section 4: Is the rarity of the event relative to the last 10000 years or to a "repetition" of the present state? I think this should be clarified because this might have implications on connections with the stability of the WAIS over the past 10K years and the initiation of the retreat in the last century.

Our frequency calculation per 10kyr is based on the CESM1 LENS pre-industrial and historical (internal component only) simulations, so the calculation reflects repeated pre-industrial climate variability. We recognize that this is an imperfect analogy to Holocene variability. Building on our response to reviewer #1, we revised the text to ensure that our results are restricted to the terminology "per 10kyr of internal climate variability" in the results, and we note the implications and caveats of this analogy to the Holocene in the discussion. In the methods (L249 of the tracked changes manuscript), we add the statement:

"The simulations allow us to quantify the rarity of the 1940s event relative to 10kyr of internal pre-industrial climate variability, providing an estimate for the significance of the event relative to the Holocene. We note that these simulations are an imperfect analogy to Holocene climate variability; we discuss the caveats associated with this analogy in the discussion section."

In the discussion (L854), we added:

"the internal components of the LENS simulations are an imperfect analogy to the Holocene, which was subject to differences in insolation, freshwater inputs, and potentially ENSO variability (e.g., Mayewski et al., 2004). However, the influence of these differences on Holocene climate variability near West Antarctica is highly uncertain. For instance, is no clear consensus on whether Holocene ENSO variability was similar (e.g., Cobb et al., 2013) or damped (e.g., Conroy et al., 2008; Grothe et al., 2020) relative to modern climate. Thus, the results presented here are associated with several caveats, but provide an estimate of the significance of the 1940s event based on the best available simulations."

**Minor comments**

- Line 30. I like this introductory paragraph and find useful to put things into context. However, I feel the wording "There is evidence that these glaciers have been relatively stable for the last ~10,000 years (Larter et al., 2014), which implies that a change in ocean circulation, and a corresponding increase in heat delivery to the ice shelves, must have occurred to initiate the current stage of retreat" to be a bit too strong here. Could a slow change in ocean forcing and/or surface mass balance be part of the story?

  We modified this statement to: "some change in ocean circulation, and a corresponding increase in heat delivery to the ice shelves, likely occurred".

  We would also like to clarify three distinct components of the narrative of glacier retreat in the ASE: (1) the historical trigger of the retreat, (2) the underlying cause of the ongoing retreat, and (3) the ongoing retreat itself. The focus of this paper is on the trigger of retreat. The ongoing retreat itself is governed by ice/ocean feedbacks (e.g., Joughin et al., 2014; Holland et al., 2023) of a greater scale than the changes from surface mass balance (Shepherd et al., 2002). We added the following text to L32 of the tracked changes manuscript to clarify this point:

  "Direct observations of glaciological, oceanic, and atmospheric conditions in the ASE from recent decades show that ice melt rates are sensitive to short-term (i.e., seasonal to interannual) changes in ocean forcing and surface mass balance (Shepherd et al., 2002, Dutrieux et al., 2014; Jenkins et al., 2018, Alley et al., 2021; Wahlin et al., 2021). However, the current stage of glacier retreat is dominated underlying ice/ocean feedbacks (Joughin et al., 2014; Holland et al., 2023). The brevity of instrumental data in this region (available from 1979 or later) makes it difficult to assess the historical trigger that initiated the current stage of glacier retreat."

- Are winds in the Amundsen Sea important only for ice shelf basal melting? What about carbon uptake, ecosystems, sea ice etc.? how a rare wind event would impact other components of the system? Might be worth adding a few lines, given that this manuscript does not focus on ice shelves.

Yes, winds in this region are likely important for many reasons. We added the following statement regarding these other motivators to the introduction at L100:

"We note that the 1940s ASE atmospheric event is likely also important for influencing other important components of the local ice/ocean system, such as carbon uptake, upwelling of nutrients, and sea ice extent (e.g., Stammerjohn et al., 2015; Yager et al., 2016;), but the primary motivation of this study is to investigate the significance of the event as a potential atmospheric trigger of glacier retreat."

- There are also other mechanisms (beyond shelf break winds) that have been proposed to potentially explain changes in ice shelf basal melting in the Amundsen Sea, including surface buoyancy forcing and remote forcing (Ross Gyre and melting of glaciers in the Western Peninsula). I would suggest to either mention that other mechanisms have been proposed and therefore this new study can provide new information on potential processes, or simplify and shorten the text highlighting that rare winds events can affect heat delivery to ice shelves and potentially their stability. Given the strong motivation and evidence based on sediments and ocean dynamical studies, the first option might be more appropriate.

Other mechanisms such as remote forcing may play a key role in driving CDW transport in the ASE, such as suggested by Nakayama et al., 2018. We refer to this in the discussion in L870, noting that it is a much less developed area of research but requires further investigation. We added an additional statement including alternative mechanisms to L857 of the discussion:

"Alternatively, recent changes in buoyancy, perhaps due to increased freshwater inputs locally or from the Antarctic Peninsula, could have played in a role in preventing ice recovery after the 1940s event (e.g., Webber et al., 2019; Flexas et al., 2022)."

- Line 130: "Magnitude" of what? Please specify.

We are referring to the magnitude of the zonal wind anomalies. We have updated that line (L172 of the tracked changes manuscript).

- Line 215: "SAT" for surface air temperature?

"SAT" and "TAS" are both commonly used for 2m air temperature, so we will keep it as "TAS".

- Line 465. I would provide a very short summary of the key results at the beginning of the Discussion.

We added a summary of the major findings of this study to the beginning of the discussion section (L916):

"In this study, we use proxy-based reconstructions of SLP and US to characterize the 1940s atmospheric anomalies over the Amundsen Sea, West Antarctica, which have previously been identified as a candidate for initiating glacier retreat in this region. The reconstructions show high pressure and westerly anomalies over the Amundsen Sea for at least five years, centered around the years 1940 and 1941. We find that the event may be a "perfect storm" of atmospheric circulation associated with the very strong El Niño event and potentially coinciding with Indian

Ocean and Atlantic Ocean variability. Our rarity calculations show that the 1940s westerly event (in terms of its 4-year mean) is expected to occur ~20 to 260 times per 10kyr of pre-industrial climate variability, suggesting that the wind event is likely not unprecedented."

- Line 500-505. The discussion here is very important as it highlights that many mechanisms are at play. As suggested before, some of these processes should be mentioned in the Introduction. Something that could be also discussed a bit more is the role of the IPO in all this. A recent study by Vance et al (https://doi.org/10.1038/s43247-022-00359-z ) suggests IPO anomalies in the 20[th] century. Would this be important for the 1938-1942 event? Or for trend/interdecadal variability over the reconstructed period?

  Holland et al. (2022) investigate the role of the IPO on atmospheric variability and century-scale trends in the ASE. The IPO is a dominant influence on interdecadal timescales, but for interannual -- which are more relevant to the 1940s wind event -- ENSO is the dominant driver and at least partly explains the anticyclonic anomalies shown in the reconstructions.

- Discussion and conclusion. Similar to the introduction, would an event like this affect other components of the system?

  It is possible that this large westerly event could have affected other components of the system, such as sea ice, biological processes and carbon uptake. Our study investigates the 1940s event as a means to test a hypothesis relating to the trigger of glacier retreat in West Antarctica; it does not directly advance our understanding of these other components, so we have decided to leave the other motivators in the introduction.

Reviewer Comment 3

This study uses paleoclimate reconstructions to assess the drivers and rarity of the 1940s events that led to large ice shelf melting and glacier retreat in the ASE. These events were rare due to their large magnitude and duration. Local forcings in combination with a major ENSO led to the 1940s events, which are rare on centennial timescales but uncommon on millennial timescales. The paper is written in an elegant way, with clear messages and figures. The subject is of interest to glaciologists and oceanographers studying West Antarctica, and I recommend publication in The Cryosphere. Below are some comments and suggestions that might help to improve the manuscript.

We thank the reviewer for their positive review and helpful comments which have improved the manuscript.

Major:

1) The authors say they evaluated the rarity of the event in the Holocene, but their simulation goes back to ~2000 yr (lines 192-194). Although this is part of the Holocene period, it does not cover the full Holocene or the Holocene conditions. I need a better explanation of why they claim the analysis covers the Holocene. Or, the authors could change Holocene to a simple "10kyr period"?

Following our response to the other reviews, we have revised the text to address the uncertainty associated with repeating the LENS simulations as an analogy for the Holocene. We have removed the

term "Holocene" from the results and clarify that the frequency calculation is an estimate of occurrences per 10kyr. We note the implications of using this calculation for an analogy to Holocene climate variability in the methods (L249 of the tracked changes manuscript):

> "The simulations allow us to quantify the rarity of the 1940s event relative to 10kyr of internal pre-industrial climate variability, providing an estimate for the significance of the event relative to the Holocene. We note that these simulations are an imperfect analogy to Holocene climate variability; we discuss the caveats associated with this analogy in the discussion section."

And in the discussion (L950):

> "the internal component of the LENS simulations are an imperfect analogy to the Holocene, which experienced differences in insolation, freshwater inputs, and potentially ENSO variability (e.g., Mayewski et al., 2004). However, the influence of these differences on climate variability near West Antarctica is highly uncertain; e.g., there is no clear consensus on whether Holocene ENSO variability was similar (e.g., Cobb et al., 2013) or damped (e.g., Conroy et al., 2008; Grothe et al., 2020) relative to modern climate. Thus, the results presented here are associated with uncertainties but provide an estimate of the significance of the 1940s event relative to the best available simulations."

2) Section 3.2: When I read "Drivers" (also in the title), I thought the authors would track down the origin of the rare 1940s event. But I have the feeling that they don't. Instead, their assumption is that other events in combination with ENSO could trigger and amplify the 1940s event. My questions remained: How this event was generated? What (local conditions) triggered this event? Saying that the 1940s event was due to local drivers and not totally ENSO-related does not properly address the "Drivers" of the 1940s event, in my opinion. I'd like to see more analysis on this to properly address what led to the 1940s event. I think this is a key question that could benefit this paper to be a greater contribution to the glaciological/ocean/atmospheric community. If the authors decided to not track down the causes of the 1940s event, which I understand can be a lot of work, I'd recommend avoiding using "Drivers". However, I think understanding the local conditions could be a great addition to the paper.

Following reviewer #1's similar comment, we added new analyses, figures, and text to investigate the potential additional drivers of the 1940s event, using the Atlantic and Indian ocean pacemaker simulations. A detailed explanation of this analysis is available in our response to reviewer #1's concern #4.

We note that this does not address the drivers of local variability in the Amundsen Sea, which have been investigated in previous studies (e.g., Raphael et al., 2016; Goyal et al., 2021). We have changed the word "Drivers" in our manuscript title to "Characteristics", and have revised our use of the word "drivers" throughout the text.

Goyal, R., Jucker, M., Gupta, A. S., England, M. H.: Generation of the Amundsen Sea Low by Antarctic orography, Geophys. Res. Lett., 48, e2020GL091487, 10.1029/2020GL091487, 2021.

Raphael., M. R., Marshall. G. J., Turner, J., Fogt, R. L., Schneider, D., Dixon, D. A., Hosking, J. S., Jones, J. M., Hobbs, W. R.: The Amundsen Sea Low: variability, change, and impact on Antarctic climate, Am. Met. Soc., 97(1), 111-121, 10.1175/BAMS-D-14-00018.1, 2016.

Minor:

- Line 294: Figure 5 instead of Figure 4?

Yes, that is a typo. We have changed the reference to Figure 5 in the revised manuscript. We thank the reviewer for catching this.

- Line 348: Figure 6e?

Yes, that is also a typo. We have changed the text in the revised manuscript.

- Why the authors use 10kyr most of the time, but sometimes 10ka?

We have changed all cases of "10ka" to "10kyr" for consistency.

- In the authors' opinion, what is most relevant for the ice shelf melting that occurred in the ASE: duration or magnitude of the 1940s event?

We suggest that it's the combination of the two (i.e., the time-integral of the anomaly) that caused a significant perturbation to the ice shelves. Modern observations and the reconstructions show that similar magnitude events have occurred several times before in only the last several decades, and similarly persistent events have occurred several times before—but never one that is both large in magnitude and duration. However, ocean simulations that investigate the sensitivity of CDW transport and ice shelf melt to wind events with different characteristics are needed to answer this.

---

## Author Response (AR2)

**Response to editor**

Dear authors,

I have received three review reports from the referees and am pleased to tell that two of them are completely satisfied with your responses to their earlier concerns. Anonymous referee #1 raises a few minor issues; however, the referee is generally satisfied with your responses. Therefore, I have judged that this paper can be published in TC after minor revision. Please prepare the final version of this paper following the comments by anonymous referee #1.

Related to the first comment by anonymous referee #1 on the usage of "Holocene", I think adding more compelling justification of the definition of "Holocene" considered in this study at the end of Sect. 2.2 is ideal. For example, if you can add a relevant reference, which conducted similar numerical simulations to investigate the "Holocene" climate, it will be enough. If you do not find any relevant reference, please consider adding more detailed explanation why the method allows us to obtain the analogous picture of the "Holocene" climate here. I also want to know why the total simulation years of 5241 (1801 + 3440) is enough for obtaining the analogous picture of the "Holocene" climate.

I look forward to receiving the revised manuscript.

Sincerely,
Masashi Niwano

Dear Dr. Niwano,

We thank you for your thoughtful evaluation of the referees' responses.

We agree with you that use of the term "Holocene" can be misleading, and we have revised the manuscript to ensure that our use of the term "Holocene" is used less often and more carefully. We have clarified what we can (and cannot) say with existing information.

The estimates presented in this study provide an estimate of the rarity of the 1940s event in the context of natural variability, but cannot address the uncertainty associated with possible changes to that variability during the Holocene. Unfortunately, sufficiently high-resolution Holocene simulations are not available. We use the ~5,241 years of the LENS simulations to conduct this calculation because they are the best simulations available; the LENS simulations have minimal wind biases in the Amundsen Sea Embayment and sufficiently high spatial and temporal resolutions. We calculate the number of occurrences of similar events in the simulations and scale that number to report the occurrences on a scale of "per 10kyr", which is not directly analogous to occurrences throughout the Holocene (which we have made sure to clarify in our revised manuscript). Please refer to our response to Referee #1 for a more detailed explanation on these points.

Although associated with uncertainty, our results greatly advance our understanding of the importance of the 1940s event as a potential trigger for glacier retreat in West Antarctica – a topic of great interest to much of the glaciology community and, we expect, many readers of *The Cryosphere*.

Thank you for your continued consideration of our manuscript.

Sincerely,

Gemma O'Connor
on behalf of coauthors

I would like to acknowledge the authors' efforts in responding to my previous comments. I appreciate the changes that have been made to make the manuscript better. However, upon my second review, some similar concerns remain in the first review to be addressed before publication.

We thank the reviewer for their constructive feedback, which we believe has improved the manuscript.

1. Upon revisiting the revised manuscript, I have to reiterate a concern I previously raised regarding the use of the term "Holocene" in the context of this study. The ensemble of pre-industrial and 20th-century experiments should not be treated as representative of the Holocene condition just because they are the same time length. Using the term "Holocene" without considering the characteristics is wrong.

We agree with the reviewer that the use of the term "Holocene" is misleading and have revised the manuscript to ensure that the term is used more carefully. We removed several instances of the term from the manuscript, which can be seen in the tracked changes manuscript. We clarify that our use of the LENS simulations provides an estimate of the rarity of the 1940s event relative to pre-industrial internal climate variability and note that this is not equivalent to a simulation of the Holocene. We further justify our use of the LENS simulations by noting that sufficiently high-resolution transient Holocene simulations are unavailable. In section 2.2, L232 of the tracked changes manuscript, we added:

"The LENS simulations allow us to quantify the rarity of the 1940s event relative to 10kyr of internal pre-industrial climate variability. We note that our calculations are not equivalent to the significance of the event relative to the Holocene, which experienced differences in variability relating to changes such as insolation and freshwater inputs. Available transient Holocene simulations are insufficient for conducting this calculation as they are only available at much lower temporal resolutions and spatial resolutions (e.g., the widely used Transient Climate Evolution of the past 21 ka (TraCE-21 ka) simulations are available only at 3.75° x 3.75° spatial resolution; He et al., 2013). The rarity calculations presented here are an imperfect analogy to the Holocene but provide a novel estimate of the significance of the 1940s event, based on the best available simulations."

The caveats are stated again in the discussion section at L597:

"Furthermore, the internal components of the LENS simulations are an imperfect analogy to the Holocene, which was subject to differences in insolation, freshwater inputs, and possibly El Niño/Southern Oscillation (ENSO) variability (e.g., Mayewski et al., 2004)."

2. The first four lines of the abstract present the research background, the following two lines articulate the study's objective, the subsequent two lines convey the findings, and the final two lines delve into the discussion. In the findings of your results, I feel it is better to say that atmospheric events in the 1940s possibly occurred about once every 100-200 years than that they are not unprecedented in 10kyr, and it may have triggered the onset of retreat in West Antarctica, together with ocean conditions. I think this way, it will be easier to understand the connection to previous studies. Linking with this comment, I suggest that the author calculate the frequency over 100 years, the 10kyr period divided by 100.

We estimate that ~17 to 250 similar events occur per 10 kyr of internal climate variability (depending on the window length, simulation used, and reconstruction used), which corresponds to .17 to 2.5 occurrences per century, or .34 to 5 occurrences per 200 years. Thus, it would be misleading to say "once every 100-200 years" because of the lower estimates from the natural-prior reconstruction. We have revised the abstract to include the estimate in units of per century and have removed the statement that our results suggest the event is not unprecedented. We have reworded the final sentence about the additional factors to improve readability. The final sentences of the abstract now read:

"Climate model simulations provide evidence that events of similar magnitude and duration may occur tens to hundreds of times per 10 kyr of internal climate variability (~0.2 to 2.5 occurrences per century). Our results suggest that the 1940s westerly event is unlikely to have been exceptional enough to be the sole explanation for the initiation of Amundsen Sea glacier retreat. Additional factors are likely needed to explain the onset of retreat in West Antarctica, such as naturally arising variability in ocean conditions prior to the 1940s or anthropogenically driven trends since the 1940s."

3. Figs 5a, 5d and 6
Can you please calculate the statistical significance from the variance and shade where it is not significant?

We calculate the variance in each pacemaker simulation ensemble mean, and stippled the areas where the variance is statistically significant with 95% confidence (where the anomaly is greater than 2 standard deviations from the mean). None of the regions on the map are significant; the maps look identical to Figures 5 and 6. This makes sense, as the 1940s event -- as constrained by the reconstructions -- is the only 2-sigma event in the 20th century, and our study finds that it is driven by a combination of sources. The pacemaker simulations are constrained by SST variability from only one tropical basin, which could explain why none of these simulations shows that the 1940s event is a statistically significant event. Instead, we revised Figure 6 by stippling areas greater than 1 standard deviation from the mean. We decided to make this revision only to Figure 6, given that 5a and 5d are simply repeated in Figure 6, with different scales.

[Figure]

**Figure 6.** Modeled SLP anomalies from 1938 to 1942 in the ensemble mean of the (a) tropical Pacific, (b) Indian Ocean, and (c) North Atlantic pacemaker simulations. Stippling is shown where anomalies are greater than 1 standard deviation from the mean (no regions contain anomalies greater than 2 standard deviations). (d-f) Same as in a-c, but for $U_S$ anomalies. Anomaly reference period is 1961-1990. We note that the color bar in this figure is smaller than that of Figure 5 due to the smaller variability in the ensemble means of the simulations (Figure 5 shows the ensemble mean and individual members from the Pacific pacemaker simulations).